# Data Assimilation using an Ensemble of Models: A hierarchical approach

Peter Rayner[1]

[1]School of Earth Sciences, University of Melbourne, Melbourne, Australia

*Correspondence to:* Peter Rayner (prayner@unimelb.edu.au)

**Abstract.** One characteristic of biogeochemical models is uncertainty about their formulation. Data assimilation should take this uncertainty into account. A common approach is to use an ensemble of models. We must assign probabilities not only to the parameters of the models but the models themselves. The method of hierarchical modelling allows us to calculate these probabilities. This paper describes the approach, develops the algebra for the most common case then applies it to the TRANSCOM intercomparison. We see that the discrimination among models is unrealistically strong, due to optimistic assumptions inherent in the underlying inversion. The weighted ensemble means and variances from the hierarchical approach are quite similar to the conventional values because the best model in the ensemble is also quite close to the ensemble mean. The approach can also be used for cross-validation in which some data is held back to test estimates obtained with the rest. We demonstrate this with a test of the TRANSCOM inversions holding back the airborne data. We see a slight decrease in the tropical sink and a notably different preferred order of models.

## 1 Introduction

Models of any interesting biogeochemical system are inexact. Either they cannot include all interesting processes, the governing equations of processes are not known exactly or computational resolution limits the accuracy of the solution. Throughout this series we stress that quantitative descriptions should be inherently statistical, meaning they must include information on the probability of any quantity, either inferred or predicted. This requires us to describe the uncertainty introduced into any quantity by that of the model. Model uncertainty is of two forms, structural and parametric. Structural uncertainties occur when we do not know the functional forms that relate the inputs and outputs of the real system or that control its evolution. In biogeochemical models these functional forms are exactly specified so that uncertainty is usually manifest as an error. Parametric errors occur when the functional forms are well-known but there is uncertainty in various quantities such as constants in physical equations, initial values or boundary conditions. Uncertainties in model predictions arising from parametric uncertainty can be generated by semi-analytic error propagation (e.g. Scholze et al., 2007; Rayner et al., 2011) or by generating ensembles of model simulations from samples of the probability density functions (PDFs) of parameters (e.g. Murphy et al., 2007; Bodman et al., 2013).

Ensemble methods dominate the study of model uncertainty. The most common approach is Model Intercomparison of which the Coupled Model Intercomparison Project (Taylor et al., 2012) for the physical climate and C[4]MIP (Friedlingstein

et al., 2006) for the global carbon cycle are prominent examples. The MIPs play a crucial but controversial role in quantifying uncertainty. First, they may underestimate uncertainty since it is impossible, even in principle, to know how well a given ensemble properly samples the manifold of possible models. On the other hand not all models are equally credible. They do more or less well at tests like fitting observations or conserving required quantities. This has led to the application of Bayesian Model Averaging (e.g. Murphy et al., 2007) in which models are tested against some criteria (such as fit to observations) and their predictions weighted accordingly.

Inverse problems or data assimilation as discussed in this volume generally treats parametric uncertainty. It uses observations and statistical inference to improve knowledge of the uncertain values (see Rayner et al., 2018, and references therein for a general introduction). Structural model uncertainty must still be included and indeed it often dominates other uncertainties. Model uncertainty is hard to characterize with analytic PDFs since errors in the functional forms will project systematically onto errors in simulated quantities. Hierarchical approaches (e.g. Cressie et al., 2009) provide a mechanism for including uncertainties in the choice of model into the formulation. For an ensemble of models this involves introducing an extra discrete variable (the index of the set of models) into the problem and calculating its probability. This probability goes under several names, e.g. the Bayes Factor (Kass and Raftery, 1995) or the Evidence (MacKay, 2003, ch.28). We can then calculate probability distributions for model parameters as weighted averages over these model probabilities. Hence this application of hierarchical Bayesian modelling is closely related to Bayesian Model Averaging (Hoeting et al., 1999; Raftery et al., 2005).

Ensemble methods are rare for biogeochemical data assimilation since there are few problems for which a useful population of assimilation systems currently exists. The clearest exception to this is the case of global scale atmospheric inversions where the TRANSCOM intercomparison (Gurney et al., 2002, 2003, 2004; Baker et al., 2006) used an ensemble of atmospheric transport models and common inversion systems to infer regional $CO_2$ fluxes from atmospheric concentrations. All these studies faced the problem of estimating properties of the ensemble such as its mean and some measure of spread. Throughout they opted for the ensemble mean and two measures of spread, the standard deviation of the maximum a posteriori (most likely) estimate from each ensemble member and the square-root of the mean of the posterior variances of the ensemble. This treated all members of the ensemble equally.

Equal weighting was challenged by Stephens et al. (2007) who compared the seasonality of vertical gradients in model simulations and observations. They found that only a subset of models produced an acceptable simulation and that this subset favoured larger tropical uptake than the ensemble mean. Pickett-Heaps et al. (2011) extended this calculation. They compared simulations using optimized fluxes with airborne profiles. This required simulating the airborne profiles using the optimal fluxes for each model. Of the four atmospheric transport models tested TM3 (Heimann and Körner, 2003) performed substantially better against this extra data than the other three.

Both the cited studies used data not included in the inversion, a procedure often called cross-validation. Cross-validation asks whether new data enhances or reduces our confidence in previous estimates while Bayesian model averaging calculates our relative confidence in two models. We shall see that the machinery needed to answer these two questions is very similar.

The outline of the paper is as follows. In Section 2 we review the necessary machinery. Section 3 describes an application to the TRANSCOM case including an extension to treat covarying model errors. Section 5 discusses the use of the machinery for

assessing cross-validation. Section 7 compares the technique with other model evaluation methods as well as discussing some computational aspects.

## 2 Theory

The following can be regarded as a development of ideas described in (Jaynes and Bretthorst, 2003, Ch.21) or (MacKay, 2003, Ch.28).

The standard data assimilation problem seeks to improve knowledge of some target variables in a model given observations. We express our knowledge as probability density functions (PDFs). The true state must be consistent with three independent pieces of information, our prior knowledge of the target variables, our knowledge of the observed quantities and the relationship between target variables and observations instantiated in an observation operator. In most applications the target variables are continuous quantities such as model parameters, initial or boundary conditions. We form the joint probability by multiplication as

$$p(\mathbf{x}, \mathbf{y} | \mathbf{x}^{\mathrm{b}}, \mathbf{y}^{\mathrm{o}}) \propto p(\mathbf{x} | \mathbf{x}^{\mathrm{b}}) \times p(\mathbf{y} | \mathbf{y}^{\mathrm{o}}) \times p(\mathbf{y} | H(\mathbf{x})) \tag{1}$$

where $\mathbf{x}$ represents the target variables, $\mathbf{y}$ the true values of the observed quantities, and $H$ represents the observation operator. Usually there is a prior or background value $\mathbf{x}^{\mathrm{b}}$ which serves as a location parameter for $p(\mathbf{x} | \mathbf{x}^{\mathrm{b}})$. Even more common is a location parameter for $p(\mathbf{y} | \mathbf{y}^{\mathrm{o}})$, usually an observed value returned by an imperfect measurement system.

We generate the final PDF for $\mathbf{x}$ by integrating over $\mathbf{y}$

$$p(\mathbf{x}) \propto \int p(\mathbf{x}, \mathbf{y}) d\mathbf{y} \tag{2}$$

In the usual case of data assimilation we only have one observation operator. Thus we often forget that the posterior PDFs for target variables are implicitly dependent on the observation operator. Where an ensemble of observation operators is available we can no longer assume certainty for which one we should use. The $i^{\mathrm{th}}$ observation operator $H_i$ becomes part of the target variables so instead of calculating $p(\mathbf{x} | \mathbf{y})$ we now seek $p(\mathbf{x}, H_i | \mathbf{y})$.[1] Once we have calculated $p(\mathbf{x}, H_i | \mathbf{y})$ we can either integrate over $\mathbf{x}$ if we are interested in the relative probabilities of different observation operators or we can sum over the various choices of observation operators to obtain the PDF for $\mathbf{x}$. The hierarchical approach (see Rayner et al., 2018, Section 5.6) factorises this joint PDF of observation operators and continuous target variables using an expression known variously as the chain rule of probabilities or the law of total probabilities. In the case of a discrete choice of observation operator this takes the form

$$p(\mathbf{x}, H_i) = p(\mathbf{x} | H_i) p(H_i) \tag{3}$$

We can apply the same rule to the joint probability in Equation 1 to yield

$$p(\mathbf{x}, H_i | \mathbf{y}) = P(\mathbf{x} | \mathbf{y}, H_i) p(H_i | \mathbf{y}) \tag{4}$$

---

[1]The true target variable is $i$, the index variable on the set of observation operators but we will continue to use $H_i$ to make it clear to what this index refers.

we see that the hierarchical and nonhierarchical PDFs differ only by the factor $p(H_i|\mathbf{y})$ and we hence need to calculate this term.

We will develop the theory for the simplest linear Gaussian case. Here many of the resulting integrals have analytic solutions. The approach will hold for nonlinear observation operators provided they are approximately linear over enough of the support for the joint distribution of $\mathbf{x}$ and $\mathbf{y}$. The qualitative ranking of models is unlikely to be sensitive to weak nonlinearities since, as we shall see, the discrimination among models is strong.

We follow the notation of Rayner et al. (2018). We switch from using a potentially nonlinear observation operator $H$ to a linear one represented by the Jacobian $\mathbf{H}$. Take a collection of linear observation operators with Jacobians $\mathbf{H}_1 \ldots \mathbf{H}_N$, with prior probability for the continuous target variables given by $G(\mathbf{x}|\mathbf{x}^b, \mathbf{B})$ and probability for the data given by $G(\mathbf{y}|\mathbf{y}^o, \mathbf{R})$ where $G(\mathbf{x}|\mu, \mathbf{C})$ represents the Gaussian distribution of the variable $\mathbf{x}$ with mean $\mu$ and uncertainty covariance $\mathbf{C}$. $\mathbf{x}^b$ is the mean of the prior distribution for $\mathbf{x}$ (often abbreviated the background value) and $\mathbf{B}$ is the prior uncertainty covariance of $\mathbf{x}$ (often abbreviated the background or prior uncertainty). $\mathbf{y}^o$ is the mean of the PDF for the observation with $\mathbf{R}$ representing the uncertainty of the observing process. See Rayner et al. (2018) Section 5 for more detailed explanation.

For each $\mathbf{H}_i$ our problem is the linear Gaussian inversion described in Rayner et al. (2018), Section 6.4. Most importantly for us the posterior PDF $p(\mathbf{x}|\mathbf{y}, \mathbf{H}_i)$ is Gaussian:

$$p(\mathbf{x}|\mathbf{y}, \mathbf{H}_i) = G(\mathbf{x}, \mathbf{x}_i^a, \mathbf{A}_i) \tag{5}$$

where $\mathbf{x}_i^a$ is the maximum a posteriori estimate for the $i^{\text{th}}$ observation operator (often called the analysis) with covariance $\mathbf{A}_i$. Substituting Equation 5 into Equation 4 we obtain

$$p(\mathbf{x}, \mathbf{H}_i|\mathbf{y}) = p(\mathbf{H}_i|\mathbf{y}) \times G(\mathbf{x}|\mathbf{x}_i^a, \mathbf{A}_i) \tag{6}$$

Thus $p(\mathbf{x}, \mathbf{H}_i|\mathbf{y})$ is a sum of Gaussian distributions.

$p(\mathbf{H}_i|\mathbf{y})$ is the marginal likelihood for a Gaussian (Michalak et al., 2005, Eq. 10):

$$p(\mathbf{H}_i|\mathbf{y}) = K \left|\mathbf{R} + \mathbf{H}_i \mathbf{B} \mathbf{H}_i^T\right|^{-1/2} \exp\left[-\frac{1}{2}(\mathbf{y}^o - \mathbf{H}_i \mathbf{x}^b)^T \cdot (\mathbf{R} + \mathbf{H}_i \mathbf{B} \mathbf{H}_i^T)^{-1} \cdot (\mathbf{y}^o - \mathbf{H}_i \mathbf{x}^b)\right] \tag{7}$$

Note that $p(\mathbf{H}_i|\mathbf{y})$ is a PDF over the indices $i$ since all terms on the rhs of Equation 7 apart from $\mathbf{H}_i$ are fixed. $K$ is chosen so that $\sum_i p(\mathbf{H}_i|\mathbf{y}) = 1$. This normalisation reflects the fact that we must choose one of the models.

## 2.1 Interpretation

One interpretation for the exponential in Equation 7 is as a ratio of the performance of the model and its expected performance. The term $\mathbf{H}_i \mathbf{x}^b - \mathbf{y}^o$ is the mismatch between the observations and the simulation using the background value. Provided $\mathbf{x}$ and $\mathbf{y}$ are independent before our assimilation, $\mathbf{R} + \mathbf{H}_i \mathbf{B} \mathbf{H}_i^T$ is the variance of this prior mismatch. This follows from the Jacobian rule of probabilities (Tarantola, 2005, Eq. 1.18) and the expression for the variance of the difference of two normally distributed quantities. Thus, inspection of the the rhs of Equation 7 shows it to be, excluding some potential normalisation, $G(\mathbf{z}, 0, \mathbf{H}_i \mathbf{B} \mathbf{H}_i^T + \mathbf{R})$ evaluated at the point $\mathbf{z} = \mathbf{H}_i \mathbf{x}^b - \mathbf{y}^o$. Smaller magnitudes of $\mathbf{H}_i \mathbf{x}^b - \mathbf{y}^o$ correspond to better a priori

simulations of the observations and higher values of $p(\mathbf{H}_i|\mathbf{y})$ i.e more likely models. Equal magnitudes of $\mathbf{H}_i\mathbf{x}^{\mathrm{b}} - \mathbf{y}^{\mathrm{o}}$ may *not* produce the same value of $p(\mathbf{H}_i|\mathbf{y})$ since The mismatch variance $\mathbf{H}_i\mathbf{B}\mathbf{H}_i^T + \mathbf{R}$ may not weight them equally. We can say equivalently that the model performance should be judged by the normalised prediction error (simulation − observation divided by its variance) penalised by the expected range of the predictions or the volume of the data space occupied by the prior model and its uncertainty (see discussion in MacKay, 2003, Ch.28).

Eq. 7 occurs in other hierarchical contexts such as the calculation of covariance parameters by Michalak et al. (2005) and Ganesan et al. (2014). This is understandable since the submodels in all three cases are the classical Gaussian problem. We note that these two papers used Eq. 7 to tune covariance parameters which may change the relative weighting of models. It raises the question that relative performance of models may depend strongly on whether the inversion is well-tuned for that model. The algorithm in Michalak et al. (2005) consists of tuning a scaling factor for prior covariances to maximize $p(\mathbf{H}_i)$ (though in their case there is only one model). We can test the sensitivity to a uniform scaling of $\mathbf{B}$ and $\mathbf{R}$ by a factor $\alpha$. Increasing $\alpha$ increases the determinant so decreases the first factor of $p(\mathbf{H}_i)$ while it decreases the negative exponent and so increases the second part. The balance is a relatively subtle change. In Section 3 we will investigate whether this is enough to change the ranking of models in one example.

Note also that for a given $\mathbf{B}$ and $\mathbf{R}$, Eq. 7 is extremely punishing on inconsistency. For example consider a case with $N$ observations and two models $\mathbf{H}_1$ and $\mathbf{H}_2$ for which the quantity $\frac{1}{N}(\mathbf{y}^{\mathrm{o}} - \mathbf{H}\mathbf{x}^{\mathrm{b}})^T \cdot (\mathbf{R} + \mathbf{H}_i\mathbf{B}\mathbf{H}_i^T)^{-1} \cdot (\mathbf{y}^{\mathrm{o}} - \mathbf{H}\mathbf{x}^{\mathrm{b}})$ (the mean square mismatch per observation) are 1.0 and 1.01 respectively. With $N = 10000$ (by no means unusually large) we see, from substitution into Equation 7 that $p(\mathbf{H}_1|\mathbf{y})/p(\mathbf{H}_2|\mathbf{y}) = e^{50} \approx 10^{22}$. This is unrealistic and is an example of the "curse of dimensionality" (Stordal et al., 2011) in which distances between points in high-dimensional spaces tend to infinity. We shall address one approach to resolving this problem in Section 4.

## 2.2 Relationship with Other Criteria

The exponent in Eq. 7 is also the minimum value of the cost function usually minimised to solve such systems. It is often denoted $\frac{1}{2}\chi^2$. In an assimilation where variances of prior PDFs and residuals calculated from the assimilation are consistent, the expected value of $\chi^2$ is equal to the number of observations (Tarantola, 1987, P.211). We often quote the normalized $\chi^2$ as $\frac{\chi^2}{n}$, roughly the mean square mismatch per observation.

$p(\mathbf{H}_i|\mathbf{y})$ is related to several other measures of model quality. For convenient comparison we define

$$L = -2\log\left(\frac{p(\mathbf{H}_i|\mathbf{y})}{K}\right) = \log\left|\mathbf{H}_i\mathbf{B}\mathbf{H}_i^T + \mathbf{R}\right| + \chi^2 \tag{8}$$

The change of sign means smaller values of $L$ correspond to more likely models.

$L$ is related to other criteria for model selection such as the Akaike Information Criterion (Akaike, 1974) and Schwartz Information Criterion (also called the Bayesian Information Criterion,BIC) (Schwarz, 1978). In our case the AIC can be defined as

$$\mathrm{AIC} = 2M + \chi^2 \tag{9}$$

where $M$ is the number of target variables (the dimension of $\mathbf{x}$). The related Bayesian or Schwartz Information Criterion is defined as

$$\text{BIC} = \chi^2 + M \ln(N) \qquad (10)$$

All three criteria consider the goodness of fit of the model. All criteria penalise models for adding parameters. Neither AIC nor BIC take account of different prior uncertainties among parameters or different sensitivities of the observations to these parameters.

## 3  The TRANSCOM Example

The **TRANSCOM** III intercomparison (Gurney et al., 2002, 2004; Baker et al., 2006) was designed to investigate the impact of uncertainty in atmospheric transport models on the determination of CO2 sources and sinks. The target variables were the mean CO2 flux from each of 22 regions (11 land and 11 ocean) for the period 1992–1996. These fluxes excluded fossil fuel emissions and a data driven estimate based on ocean and atmosphere measurements (Takahashi et al., 1999). Prior estimates and uncertainties were gathered from consultation with experts in each domain. The data was the average CO2 concentration from 76 stations and the same data was used in every inversion. Participants in the intercomparison calculated Jacobians by inserting a unit flux into an atmospheric transport model corresponding to each region. There were 17 participating models so our space of target variables consists of 22 flux components and an indexed set of 17 models $\mathbf{H}_i$.

The inversions for the flux components are carried out by changing $\mathbf{H}$ with all other aspects held constant. The authors then created pooled estimates of the posterior fluxes such as the mean, the mean uncertainty (averaging all the posterior uncertainties) and finally the "between model" spread, calculated as the covariance among the posterior fluxes for each model. In all these calculations we weighted every model equally. What happens if we apply the methods described in Section 2 to calculate pooled estimates?

Figure 1 shows a slightly modified $L$ for the seventeen models for the cases without (top) and with (middle) tuning following Michalak et al. (2005). The modification consists of displaying $\log_{10}$ rather than the natural logarithm. For the tuning cases we used one multiplier each for $\mathbf{B}$ and $\mathbf{R}$. We see a large range of weights, 11 orders of magnitude for the untuned and 14 orders of magnitude for the tuned cases. This certainly reflects the "curse of dimensionality" mentioned earlier. For the same reason there is a strong focus of weight on a few models. Tuning intensifies this focus though it leaves the ranking almost unchanged. We conclude therefore that variation in model performance (as measured by $L$) does not reflect the quality of tuning of the inversion but something more fundamental about the models and data. Henceforth we consider only the untuned case.

Although rankings do not change much, we see that model 3 and, to a lesser extent, model 1 have much lower weights after tuning than before. The variance tuning procedure reduces the variances for most models (indicating that they fit the data better than the original variances suggest they should). All else being equal a lower optimal value for the variance scaling factors means an increased $p(\mathbf{H}_i|\mathbf{y})$. Models one and three do not have their variance scaling changed much so their relative weight is reduced. The reduction is large because of the same dimensionality arguments made above.

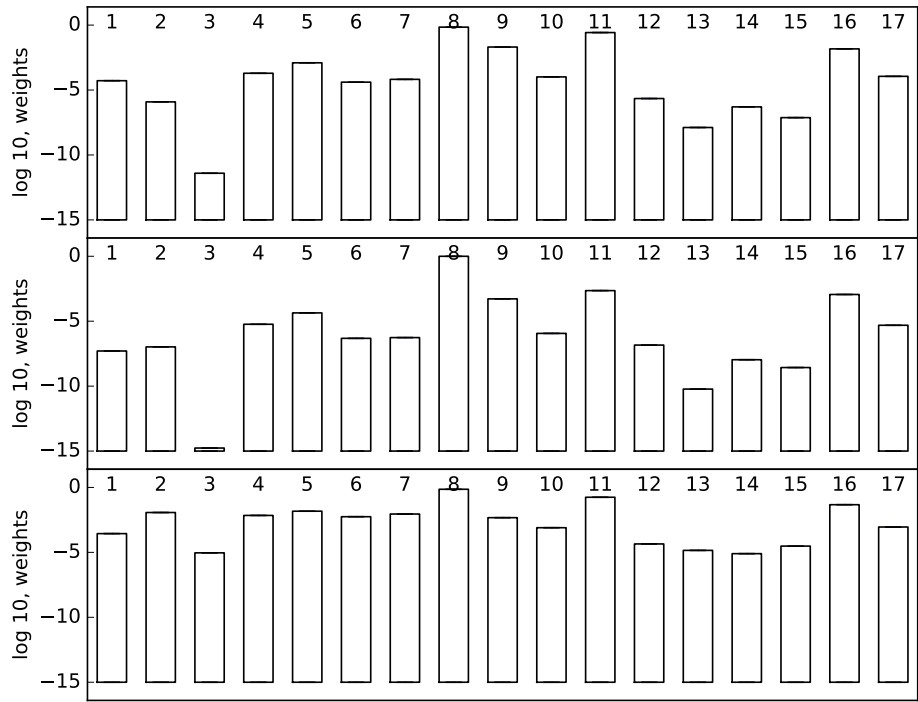

**Figure 1.** $\log_{10}$ of $p(\mathbf{H}_i|\mathbf{y})$ for the untuned (top), tuned (middle) and case with residuals used for $\mathbf{R}$ (bottom) transcom inversions.

In the next two sections we consider the marginal probabilities to investigate the relative probabilities of different models and the pooled flux estimates.

## 3.1 Model Probabilities: Comparing Model Performance

The Gaussian weights derived in Section 2 are the probabilities that a given model is the correct one for matching the data under the assumption that we must choose one (see Jaynes and Bretthorst, 2003, p.136 for a discussion of this point). We must, however, be careful not to over-interpret these probabilities as measures of model quality. In the first place, $L$, like the BIC and $\chi^2$ grows with the number of observations. So, then, does the divergence among models, an effect intensified when we take exponentials to calculate probabilities. The relative quality of two models depends on the amount of data used to compare them even if our ability to distinguish between them does increase as we add data. We can normalise by considering $L/N$ (where $N$ is the number of observations) as a generalisation of the normalised $\chi^2$. This ranges from a minimum of 0.01 to 0.67. The very low value should not be interpreted as representing an absolute quality of fit since we have normalised the probabilities to sum to 1. Rather it tells us that the apparently large change in the weights is a result of much smaller differences in the relative quality of the fit coupled to large amounts of data.

## 3.2 Ensemble Means and Variances

Once we sum over $i$ we obtain $p(\mathbf{x}|\mathbf{y})$ as a sum of Gaussian distributions with fixed weights. These are usually referred to as Gaussian mixture distributions. We can calculate various statistics of the ensemble using well-known properties of Gaussian mixtures. The mean is calculated as

$$\mu = \sum_i p(\mathbf{H}_i|\mathbf{y})\mathbf{x}_i^{\mathrm{a}} \tag{11}$$

Note that this collapses to the conventional mean if all weights are equal. The variance is calculated as

$$\mathbf{A}^* = \sum_i p(\mathbf{H}_i|\mathbf{y})\left[\mathbf{A}_i^* + (\mathbf{x}_i^{\mathrm{a}} - \mu)^2\right] \tag{12}$$

The superscript $*$ indicates we consider only the diagonal of the relevant matrices; Equation 12 only accounts for the variance not the covariance of the estimates. The second term in Equation 12 includes the spread of the means for each model. If all the $p(\mathbf{H}_i|\mathbf{y})$ are equal, Equation 12 collapses to the "total uncertainty" metric used by Rayner (2004) to incorporate both the "within" and "between" model uncertainty described in Gurney et al. (2002).

Figure 2 shows the equally-weighted and probability-weighted case for the **TRANSCOM** regions, in a format following Gurney et al. (2002). Here we do not show the "within" and "between" metrics separately since the Gaussian mixture naturally combines them. The focus of $p(\mathbf{H}_i|\mathbf{y})$ on a few models (70% on one model) might suggest that the uncertainty in the weighted case should be far smaller than the equally weighted traditional case. Figure 2 shows this is not the case. Both the means and uncertainties for the two cases are quite similar.

The agreement of the means is explained by a result from Gurney et al. (2002). They noted that the mean simulation from their equally-weighted ensemble produces a better match to the data than any individual model . The probability-weighted flux is constructed to maximize the posterior probability across the model ensemble and parameter PDFs thus its mean should also produce a good match. It is hence no surprise that the preferred model eight is the model closest to the unweighted model mean. Recalling that the ensemble weights this preferred model at 70% we see good agreement between weighted and unweighted means.

The similarity in the weighted and unweighted total uncertainty is partly a result of the weak data constraint in our problem. Gurney et al. (2002) noted that for almost all regions the "within" uncertainty was larger than the "between". Furthermore the posterior uncertainties produced by each model are rather similar so that the weighted and unweighted contributions in equation 12 are similar. The contributions of the "between" uncertainty are different in the weighted and unweighted cases but, since these are smaller than the other contribution, we do not see a large final difference. This would change in cases where the constraint afforded by the data (as evidenced by the uncertainty reduction cf the prior) was large.

## 4 Improved Treatment of Observational Covariance

Although mathematically correct, the strong discrimination among models by $L$ is not intuitively reasonable. One reason for the strength of the discrimination is that each datapoint makes an independent contribution to the PDF. This is not an error in

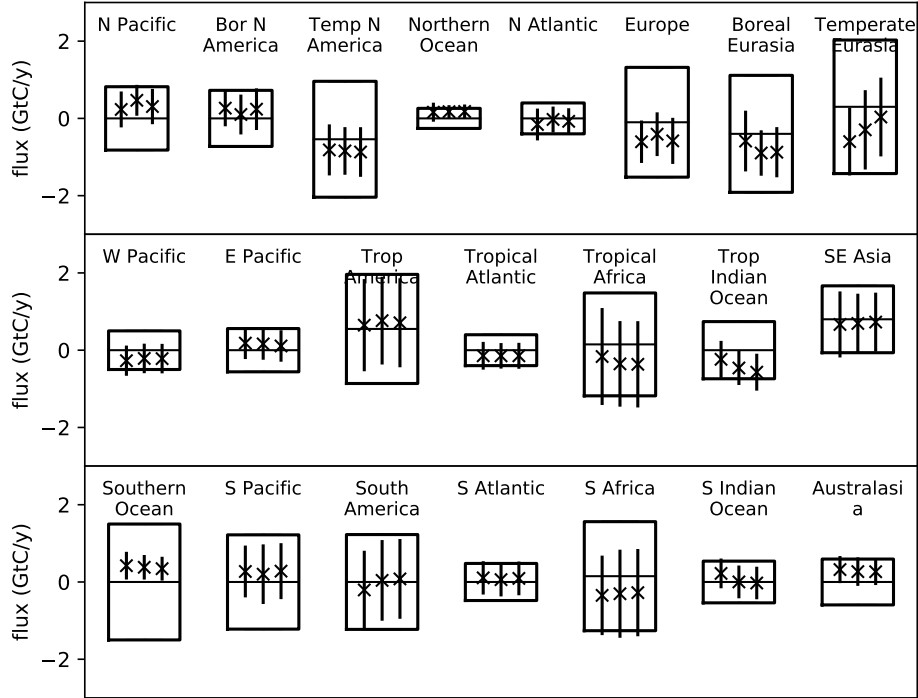

**Figure 2.** Prior and posterior uncertainties for regional fluxes from the TRANSCOM intercomparison following Gurney et al. (2002). The centre line of each box shows the prior estimate of the mean while the box limits show the $\pm 1\sigma$ uncertainties. The three bars show the mean (marked with "x") and $\pm 1\sigma$ uncertainty denoted by the length of the bar. The uncertainty is that of the ensemble including both the uncertainty for each model and the dispersion among model means. The left bar shows the equally weighted case, the middle bar the case for the $p(\mathbf{H}_i|\mathbf{y})$ and the right bar the case with covariance of residuals included.

the formulation of $L$ but rather the PDF associated with the data in the underlying assimilation.[2] In the case of atmospheric transport models this assumption says that if a model makes an error at one station, one cannot assume it will make a similar error at a nearby station. The physical coherence of atmospheric transport processes makes this most unlikely, even if subgrid heterogeneity lends some independence to the two stations.

5   There are two major approaches to characterising the model error covariance, either a priori or a posteriori. A priori we would like some machinery for calculating how uncertainties in model components or drivers project into model simulations. Lauvaux et al. (2009), for example, described a mechanism for calculating correlations in simulated tracer distributions due to correlated meteorological uncertainty but this is not a comprehensive description, i.e it leaves out many sources of uncertainty. If we have an ensemble of models we can use the ensemble of simulations using the prior value of the target variables as a

10   measure of the model contribution to uncertainty. This was suggested by Tarantola (1987). The motivating argument is that the ensemble of models samples the uncertainty of the observation operator while maintaining physical consistency for each

---

[2]Strictly speaking it is the model PDF from Rayner et al. (2018), but we have combined model and data uncertainties following their Section 6.4

member of the ensemble. Equation 1 requires the PDF of the simulation $H(\mathbf{x})$ for any $\mathbf{x}$. Tarantola (1987) suggests that the covariance of this PDF can be calculated using $\mathbf{x}^{\mathrm{b}}$.

First define the model mean

$$\mu^{\mathrm{b}} = \overline{\mathbf{H}\mathbf{x}^{\mathrm{b}}} \tag{13}$$

where the average is taken over the ensemble of models. We can then write the ensemble covariance as

$$\mathbf{R}_{i,j}^{\mathrm{prior}} = \overline{(\mathbf{H}\mathbf{x}_i^{\mathrm{b}} - \mu_i^{\mathrm{b}})(\mathbf{H}\mathbf{x}_j^{\mathrm{b}} - \mu_j^{\mathrm{b}})^{\mathrm{T}}} \tag{14}$$

where once again the average is over the ensemble of models and the subscripts index the observations.

The second approach is analysis of the posterior residuals. Desroziers et al. (2005) noted that the residuals must be consistent with the PDF assumed for the model-data mismatch, here described by $\mathbf{R}$. If this is not the case we need to make a correction
to $\mathbf{R}$. Here again we have a range of choices. If we have enough data we can fit covariance models as functions of space and time. We do not have enough data so we calculate directly the ensemble covariance of the residuals as

$$\mathbf{R}_{i,j}^{\mathrm{sample}} = \overline{(\mathbf{H}\mathbf{x}_i^{\mathrm{a}} - \mathbf{y}_i)(\mathbf{H}\mathbf{x}_j - \mathbf{y}_j)^{\mathrm{T}}} \tag{15}$$

where the overbar denotes an average over the ensemble of models and their respective analyses and the indices $i$ and $j$ refer to observations. Descriptively $\mathbf{R}^{\mathrm{sample}}$ will be positive if, on average, models make errors of the same sign for observations $i$
and $j$. Note that if the ensemble of models is smaller than the number of observations (usually the case) then both $\mathbf{R}^{\mathrm{sample}}$ and $\mathbf{R}^{\mathrm{prior}}$ are singular. Neither $\mathbf{R}^{\mathrm{prior}}$ nor $\mathbf{R}^{\mathrm{sample}}$ capture observational error however. (Tarantola, 2005, Eq. 1.106) points out that, for Gaussian PDFs, we can combine the PDFs for the model and observations by adding their respective covariances.

We note in advance an objection to using $\mathbf{R}^{\mathrm{sample}}$ that, by using the residuals, we are double counting information in any subsequent inversion. This is partly true although firstly we only use it to correct the spread not the location of the related PDFs
and that the same objection holds for any use of posterior diagnostics.

Figure 3 shows, for a sample of stations, the first-guess and residual standard deviations from Eq. 14 and Eq. 15 as well as the control standard deviation. The standard deviations show somewhat similar structure, with the largest values for terrestrially-influenced stations such as Baltic Sea (bal), Hungary(hun) and Taiane Peninsula, Korea (tap). Magnitudes of the first-guess variances are larger however. There are 49 of the 76 stations where the first-guess variance is larger than the observational error
variance but only one station (Mould Bay in Canada) where this is true for the residual variance. This reflects the convergence of simulations towards the observations. Covariances among stations are more complex but, as expected, they are strong whenever multiple measurements occur near each other (e.g. at one station or in one vertical profile). This has the desirable property of de-weighting these measurements relative to an independent observational covariance.

The weights for the case using $\mathbf{R}^{\mathrm{sample}}$ is shown as the bottom row in Figure 1 and the impact on regional estimates is shown
as the right bar in Figure 2. The ranking is similar to the other cases, especially for the preferred models. The main effect of including the residual covariance is to reduce the penalty for the least preferred models. Given the small changes among the preferred models it is no surprise that there is little change in the regional estimates or total uncertainties. One reason for the

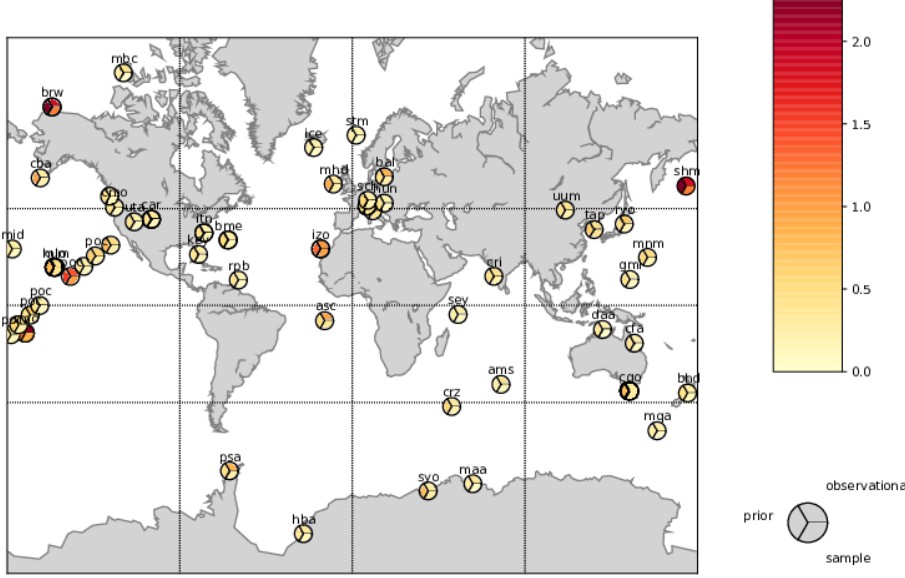

**Figure 3.** Assumed standard deviations, standard deviations taken from the diagonal of $\mathbf{R}^{\mathrm{prior}}$ and $\mathbf{R}^{\mathrm{sample}}$ for a representative subset of stations.

largest impact falling on the least preferred models is that the residual covariance is dominated by the largest residuals which come from the least preferred models.

## 5 Model Comparison and Cross-Validation

In Section 3 we applied the theory to the simplest possible case of models with identical dimensionality and uncertainties; they differed only in their Green's Function. The theory is more general than this. We noted in Section 2.1 that model performance is determined by the normalised prediction error and the volume of the data space occupied by the prior model. Neither of these depends directly on the dimensionality of the prior model. We can compare a model with two highly uncertain parameters against another with four more certain parameters. This extends the BIC which considers only the number of parameters. The case is quite common in biogeochemistry in which we often compare simple models with empirical and highly uncertain parameters with complex, physically-based models whose parameters can be linked to field experiments.

A special case occurs when we compare the prior and posterior models. This is usually done by holding back a subset of the data and testing the improvement in the fit to that data (e.g. Peylin et al., 2016). The approach is frequently called cross-validation. $L$ provides a good basis for comparison of the prior and posterior models. Most importantly it accounts for the different volumes in the data space occupied by the prior and posterior models. Posterior models (informed by the previous assimilation) always occupy less volume in the space of the cross-validation data than their unconstrained or free-running prior model. Thus a good fit to the cross-validation data is less likely to be a chance event.

It is also possible to weight model estimates by their ability to fit cross-validation data. The steps are as follows:

1. Divide data into assimilation and validation data;

2. Carry out an ensemble of assimilations using each model and the assimilation data;

3. Calculate $L$ using the *posterior* estimates from step two and the validation data;

4. Calculate ensemble statistics from the posterior estimates from step two and $L$ from step three.

Note that the prior means and covariances in Equation 7 for step three are the posterior means and covariances from step two. Thus, while in Section 3.1 we varied only the model $\mathbf{H}$ here we also vary $x^{\mathrm{b}}$ and $\mathbf{B}$. Variations in $\mathbf{B}$ or, more generally, variations in the projection of prior uncertainty into observation space are not usually treated in cross-validation studies (e.g. Pickett-Heaps et al., 2011).

For our example we parallel the test of Stephens et al. (2007). They held back data from airborne profiles and rated models according to their ability to fit seasonal changes in vertical gradients. We cannot use the same measure in our annual mean experiment but we do use the nine points from the airborne profiles above Cape Grim Tasmania or Colorado USA.

We can calculate $L$ using these nine measurements and the prior and posterior models. The comparison of $L$ for these cases shows whether the fit to the data held back from the inversion has improved. One would hope so but Peylin et al. (2016) showed that this is not always the case. In our case $L$ improves by several orders of magnitude due both to a reduction in the residuals

and a narrowing of the PDF. Figure 4 shows the comparison of normalised $L$ for the prior (top) and posterior (bottom) models. The prior case shows little variation around the equally-weighted value of $\frac{1}{17}$ while this variation is considerably increased for the posterior case. Figure 5 shows the ensemble statistics for three inversion cases. The left bar is the equally weighted case for the entire network (the left bar from Figure 2), the middle bar shows the equally weighted case for the inversion with

the nine cross-validation stations removed while the right bar shows the same inversion but weighted according to $p(\mathbf{H}_i|\mathbf{y}^{\mathrm{cv}})$ where $\mathbf{y}^{\mathrm{cv}}$ is the cross-validation data. Averaged across all regions the impact of changing network and changing weighting are comparable although the largest changes are in North and South America following from the change of network. This was also observed by Pickett-Heaps et al. (2011).

## 6    Computational Aspects

The hardest part of the calculation of $p(\mathbf{H}_i|\mathbf{y})$ is calculating the matrix $\mathbf{H}_i\mathbf{B}\mathbf{H}_i^T+\mathbf{R}$. There are several possible routes depending on the size of the problem and the available machinery. In problems with few parameters it may be possible to calculate and store $\mathbf{H}_i$ directly. Recall that $\mathbf{H}_i = \nabla_{\mathbf{x}}\mathbf{y}$. We can calculate $\mathbf{H}_i$ either as the tangent linear of $H$ (Griewank, 2000) or via finite difference calculations in which a parameter is perturbed. Once we calculate $\mathbf{H}$ we can generate the eigen-values of $\mathbf{H}_i\mathbf{B}\mathbf{H}_i^T+\mathbf{R}$ from the singular values of $\mathbf{H}_i$. In other cases $\mathbf{R}$ is sparse in which case we can calculate $\left(\mathbf{H}_i\mathbf{B}\mathbf{H}_i^T+\mathbf{R}\right)^{-1}$ as

a correction to $\mathbf{R}^{-1}$ using the ShermanMorrisonWoodbury formula (Cressie and Johannesson, 2008).

If the problem is too large or the generation of the Jacobian too costly we need to generate an approximation of the determinant of $\mathbf{H}_i\mathbf{B}\mathbf{H}_i^T+\mathbf{R}$. A common approach is to calculate the leading eigenvalues of (the symmetric matrix $\mathbf{H}_i\mathbf{B}\mathbf{H}_i^T$

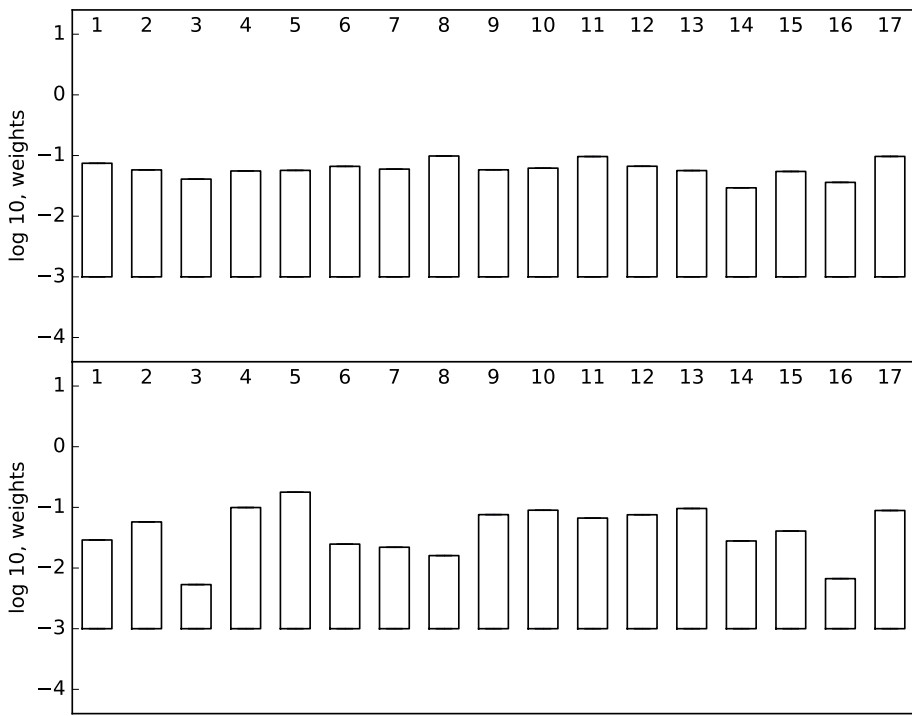

**Figure 4.** $\log_{10}$ of $p(\mathbf{H}_i|\mathbf{y})$ for the prior (top) posterior (bottom) with $p(\mathbf{H}_i|\mathbf{y})$ calculated using nine airborne measurements over Cape Grim and Colorado.

through a so-called matrix-free approach. Rather than an explicit representation of the matrix, matrix-free approaches require the capability to evaluate the product of the matrix in question with any given vector. The prime example of a matrix free approach was published by Lanczos (1950). In our case the application of a matrix-free approach requires the tangent linear of $H_i$ to generate $\mathbf{H}_i\mathbf{x}$ and the adjoint model to generate $\mathbf{H}_i^{\mathrm{t}}\mathbf{x}$. This is similar to calculations performed in the conjugate gradient algorithm for the assimilation problem itself (Fisher, 1998). The second term in Equation 7 is the Bayesian least squares cost function evaluated at the minimum so, provided we want to calculate $x^{\mathrm{a}}$ and not just $p(\mathbf{H}_i|\mathbf{y})$, we already have this value.

## 7 Discussion and Future Work

The method we have outlined points out one way of incorporating measures of model quality into ensemble estimates. The TRANSCOM case points out its main limitation, a strong dependence on the underlying PDFs. The same limitation holds for other calculations with the underlying PDFs, especially measures of information content or posterior uncertainty. Thus the largest effort needed to improve our calculation is the same as that for many other aspects of assimilation, namely the assessment of the independent information available from large sets of observations, accounting for systematic errors in observation operators. This problem is particularly difficult in biogeochemical assimilation. The normal application is of a single assim-

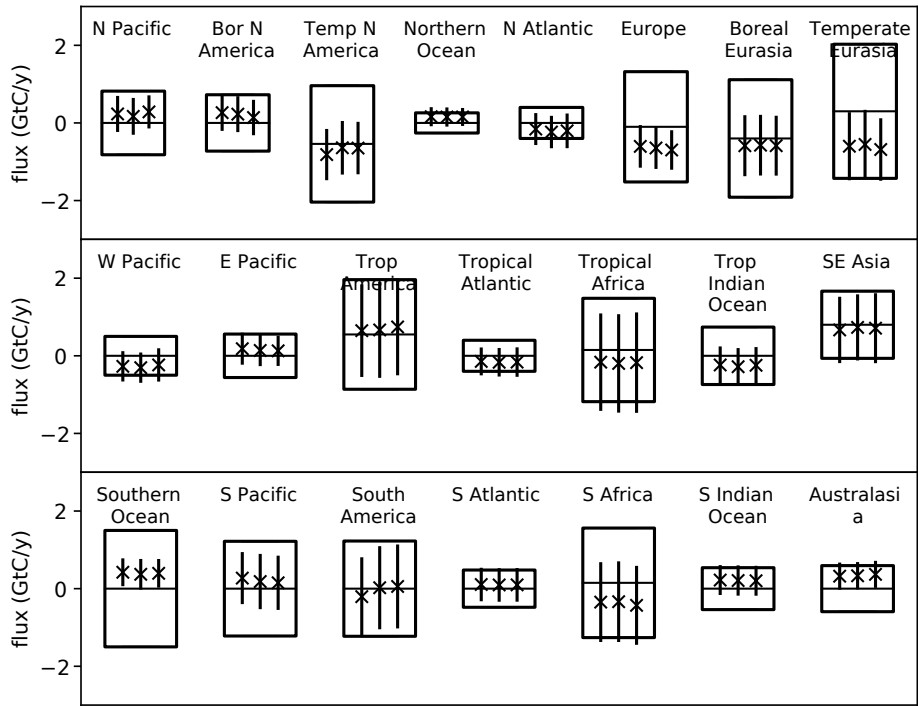

**Figure 5.** Prior and posterior uncertainties for regional fluxes from the TRANSCOM intercomparison following Gurney et al. (2002). The centre line of each box shows the prior estimate of the mean while the box limits show the $\pm 1\sigma$ uncertainties. The three bars show the mean (marked with "x") and $\pm 1\sigma$ uncertainty denoted by the length of the bar. The uncertainty is that of the ensemble including both the uncertainty for each model and the dispersion among model means. The left bar shows the equally weighted case for the full network, the middle bar the equally weighted case with the cross-validation stations removed and the right bar the $L$-weighted case for the cross-validation data.

ilation carried out over the longest possible period. This is desirable both because there is usually little data available in any period (encouraging maximising the assimilation window) and many of the processes we seek to elucidate are slow so that long windows are desirable to reveal them. This means that it is hard to separate systematic errors arising from the prior, the data itself or the observation operator.

5    Some assimilation problems are less subject to this weakness. In numerical weather prediction, for example, we have repeat assimilations. Thus we can test that the underlying PDFs are consistent with their realisations. We also have more direct tests of the quality of the assimilation via forecast skill. The above argument suggests a strong need for ensemble approaches in biogeochemical assimilation.

A more immediate application than properly weighting an ensemble of models may be in model development. Here a
10   common question is of complexity over simplicity. If, as is argued throughout this series, assimilation is a good guide to parameter choice and even structure in models we need some way to tell whether adding extra processes, with their concomitant uncertainties, is worth the effort. This is a standard problem in statistical inference. The Bayesian formulation outlined here

shifts the comparison of two models from complexity to the volume of data space available to them, allowing both complexity and uncertainty to play a role. This offers a promising basis for comparing different versions of a model.

The comparison between models and data sets is, however, incomplete. We cannot compare easily two assimilations with different amounts of data since $p(\mathbf{H}|\mathbf{y})$ has a strong dependence on dimension.

## 8   Conclusions

We have developed a simple application of hierarchical data assimilation to incorporate choice among an ensemble of models. We have demonstrated it for a computationally simple case, the annual mean version of the TRANSCOM intercomparison. The method provides unrealistically strong discrimination among models, mainly due to incorrect assumptions about underlying PDFs. We have also successfully applied the technique to the cross-validation of the TRANSCOM inversions by holding back airborne data over Tasmania and Colorado. The method, when coupled with more sophisticated diagnostics of model-data mismatch should prove a useful extension to traditional biogeochemical data assimilation.

### Code and Data Availability

The code and data files to run the TRANSCOM example and generate the figures in the paper can be found at https://figshare. com/articles/Code_needed_to_run_the_transcom_ensemble_weighted_probability_case_for_Data_Assimilation_using_an_Ensemble_ of_Models_A_hierarchical_approach_Geoscience_Model_Development_Discussions_2016_w_draft_item/4210212

*Acknowledgements.* this work was partly supported by an Australian Professorial Fellowship (DP1096309). We acknowledge the support from the International Space Science Institute (ISSI). This publication is an outcome of the ISSI's Working Group on "Carbon Cycle Data Assimilation: How to Consistently Assimilate Multiple Data Streams".

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
