# Peer review of "Data Assimilation using an Ensemble of Models: A hierarchical approach"

_Atmospheric Chemistry and Physics, 2017_

## Referee Comment (RC1) · A. Braverman (Referee) · 11 Mar 2017

Referee Report for
*Data Assimilation using an Ensemble of Models:*
*A hierarchical approach*

**General comments:**

In general I like this paper a lot. However, I find it extremely difficult to follow because of some type-o's and much notation with which I am not familiar. The notation seems inconsistent in distinguishing between fixed quantities and random ones, and indicating
where conditioning has taken place. I admit that I come from a different community, and at least some of my comments may be due simply to this notational difference. My comments below pertain to only the first part of the paper as I need these items clarified in order to proceed. Thanks for your patience in answering these questions.

**Specific comments for the authors:**

Section 2, through Section 2.1

1. Page 3, lines 6 and 7: Please define the random variables $x$ and $H_i$. In what sense is $P(x|H_i)$ "the conventional data assimilation problem"?

2. Page 3, line 9: To what "linear model" are you referring? A linear transport model represented by $H_i$? What do you mean by "over enough of the relevant pdf's"? Or, do you mean "over enough of the support of the random variable $H_i$?

3. Page 3, line 12 and 13: Is $\mathbf{H}_i$ the same as $H_i$? $\mathbf{H}_1, \ldots, \mathbf{H}_N$ are defined here as Jacobian matrices corresponding to $N$ different transport models "...with unknowns defined by the multivariate Gaussian $G(\mathbf{x}^b, B)$...". Which unknowns?

4. Page 3, line 15: "For each $\mathbf{H}_i$ our problem is the simple linear Gaussian inversion..." What does this mean? What is it you are trying to solve for or infer? Is it the flux that gave rise to the observed concentrations?

5. Page 3, line 16: "Most importantly for us $P(x^a|\mathbf{H}_i)$ is Gaussian." Please define $x^a$. Should it be $\mathbf{x}^a$?

6. Page 3, lines 16 and 17: $P(\mathbf{x}, \mathbf{H}_i)$ appears to be a joint distribution of two quantities: the vector-valued $\mathbf{x}$ and the matrix-valued $\mathbf{H}_i$. It's unclear from the notation

whether $\mathbf{H}_i$ is a random matrix or a fixed matrix. (On line 21, $\mathbf{H}_i$ is treated as random.) My guess is that it is fixed since the right side of the equal sign appears to show the pdf of just one variable; presumably $\mathbf{x}$. Is $\boldsymbol{\mu}_i$ a vector or a scalar? Please define $\mu_i$, $\mathbf{U}_i$, and $W_i$. The expression $P(\mathbf{x}, \mathbf{H}_i) = W_i G(\mu_i, \mathbf{U}_i)$ does not define a proper pdf unless $W_i = 1$ since the area under the pdf must equal one. A more precise definition of a mixture would be in terms of random variables: $\mathbf{X} = \sum_{i=1}^{K} A_i \mathbf{X}_i$, $A_i = \begin{cases} 1 \text{ with probability } w_i, \\ 0 \text{ otherwise} \end{cases}$, $\sum_{i=1}^{K} w_i = 1$, and $\mathbf{X}_i \sim G(\boldsymbol{\mu}_i, \mathbf{U}_i)$.

7. Page 3, line 23: Either $\mathbf{x}$ should be bold, or not. Do not mix within the same equation. Also, the notation $G(\boldsymbol{\mu}_i, \mathbf{U}_i)(\mathbf{x})$ seems is very confusing (to me, at least). Do you mean that $\mathbf{x}$ is an argument to the function $G$? Why not write $G(\mathbf{x}|\boldsymbol{\mu}_i, \mathbf{U}_i)$?

8. Page 3, line 26: In this equation $\mathbf{H}_i$ is treated as a non-random quantity. Above in line 21 it was random. Have you conditioned on it? If so, this distribution should be written as a conditional distribution. If not, then $W_i$ is a random variable, not a fixed weight.

9. Page 3, line 26: I can't check this equation because I can't follow the derivation in Appendix A. See below.

10. Page 3, line 27: I think there is an extra "v" at the end of this line.

11. Page 4, line 2: I assume that $\mathbf{x}_b$ has a prior distribution somewhere because it is being treated as both random and fixed in various places. What is the prior distribution?

12. Page 4, line 16: Type-o.

13. Page 4, Footnote is missing.

Appendix A, through page 12

1. Page 12, line 18: $K$ was defined in the main text as a normalizing constant. What is $K(\mathbf{H}_i)$ here? Do you mean $P(\mathbf{H}_i)$?

2. Page 12, line 21: I am confused by this equation. $G$ is a function that has an argument and parameters. What are the parameters and what are the arguments in this expression? The definitions from Section 2, lines 13 and 14 should be restated here and clarified as indicated earlier.

3. Page 12, line 23: Please define $\sigma$. Why is $\mathbf{x}$ in bold while $dx$ is not?

4. Page 12, line 25: I find the use of $\mathbf{H}_i$ as both a Jacobian and an indicator of model identity to be very confusing. Why not let $\mathbf{H}_i$ be the Jacobian of model $i$, and introduce a model indicator, say $\Delta$, an integer-value random variable taking values in $1, 2, \ldots, M$, where $M$ is the number of models?

---

## Referee Comment (RC2) · Anonymous Referee #2 · 26 Apr 2017

Summary of review

The article describes Bayesian model averaging for combining inferences from multiple flux inversion experiments. The approach seems reasonable, although I have some reservations on some of the steps taken, which I detail below. I hope the author finds the comments helpful.

General Comments

- The Introduction (Section 1) is well-presented. I particularly agree with the comment "the posterior PDFs for unknowns are implicitly dependent on the choice of model," a fact commonly overlooked.

- It is important to note that all the theory from Section 2 is conditional on the same data $y$ being used in all of the inversions. Is this the case in the Gurney study? What can be done when $y$ differs across models?

- I disagree with the introduction of JIC. Isn't this just the marginal log-likelihood of the $i^{th}$ component? I think it should be described as such. Also it is well-known that the marginal log-likelihood penalises for complex models and the marginal likelihood is commonly employed in model selection.

- Are you sure that if we replace $H_i B H_i^T + R$ with $I$ we retrieve the BIC? The BIC plugs-in maximum likelihood estimates of $x$ into the log-likelihood, while the marginal likelihood (which you label JIC) integrates out $x$, which is different.

- P3 L30: The comment 'We don't believe that the relative quality of two model depends on the amount of data used to compare then...' is slightly out of place. There are many texts that show that asmyptotically these criteria hold (under some assumptions). If there is reason to believe that the assumptions are not valid (and the criteria are hence not valid) for the flux inversion problem discussed, then some other theoretical foundation for the use of a different criterion is needed.

- Related to the previous comment, the theory shows that the posterior flux is a weighted Gaussian mixture with weights $\{W_i\}$. What is the theoretical justification of using the marginal log-likelihood to weight when combining (as I think is implied when using the expression 'JIC-weighted')? The $\{W_i\}$ being degenerate is unfortunate but not a valid reason. The same comment holds for the cross-validation metrics.

- P8 L25: I cannot find a reasonable justification as to why one should add $R$ and $R^{sample}$. There are justifiable alternatives – for example in spatial analysis one would fit a space-time model to the residuals with nugget and use this for the

new $\boldsymbol{R}$. This will be guaranteed to be non-singluar and positive definite. I think that adding these two matrices can lead to unforeseen consequences in other settings.

Other Comments

- P1 L24: I do not agree with the comment 'When structural uncertainty enters the problem only ensemble methods are available'. Although less direct, cannot one introduce an additional unstructured residual term in the model and see if that dominates?

- Section 2 and Appendix A need work to make the notation consistent. Just some things I picked up:

  - $H_i$ should be bold everywhere
  - The lack of conditioning on the data $\boldsymbol{y}$ in all equations makes it hard to distinguish between prior and posterior distributions. Also the use of $K(\boldsymbol{H}_i)$ as prior is confusing since $K$ is a normalising constant in Equation (6).
  - The matrix $\boldsymbol{B}$ in Section 2 has become $\boldsymbol{P}$ in Appendix A.
  - In Appendix A, the PDF $G(\cdot)$ takes three arguments, while in Section 2 it only takes 2.
  - Appendix A needs to be cleaned up, there are expressions like 'mathbf' appearing, $\mu$ is not bold, the subscript $i$ could be apparent on the LHS but not on the RHS, etc. See also my comment on clarifying the maths in Section A further below.

- P3 L27: Instead of "assumption that we must choose one" I would instead say "assumption that the sum of the probabilities of the models given the data equals one." Strictly speaking, unless you are using a Bayesian estimator you do not need to choose any specific one model.

- P4 L7: The statement 'Eq.6 is also the same expression as the maximum likelihood estimate in Michalak et al. (2005)' is inaccurate. Estimate of what? Eq.6 is the marginal likelihood of the data under the $i^{th}$ component.

- P5 L32: What is the difference between $JIC/N$ and $JIC$ if $N$ is constant? Does scaling affect any of the results and conclusions?

- P7 L9: Why 'model seven'? From Figure 1 it looks like Model 8 is the best model?

- P12: I found that the maths from Equation (A10) onwards becomes a bit obscure. The result is correct, however I think more details are needed. Using the author's notation, first the exponents in (A10) should be 1/2 so that

$$\mathsf{p}(\boldsymbol{H}_i) \propto |\boldsymbol{A}\boldsymbol{P}^{-1}\boldsymbol{R}^{-1}|^{1/2} \exp\left(-\tfrac{1}{2}(\boldsymbol{y}-\boldsymbol{H}_i\boldsymbol{x}^b)^T \cdot (\boldsymbol{R}+\boldsymbol{H}_i\boldsymbol{P}\boldsymbol{H}_i^T)^{-1} \cdot (\boldsymbol{y}-\boldsymbol{H}_i\boldsymbol{x}^b)\right)$$

$\propto |\boldsymbol{A}|^{1/2}\exp\left(-\tfrac{1}{2}(\boldsymbol{y}-\boldsymbol{H}_i\boldsymbol{x}^b)^T \cdot (\boldsymbol{R}+\boldsymbol{H}_i\boldsymbol{P}\boldsymbol{H}_i^T)^{-1} \cdot (\boldsymbol{y}-\boldsymbol{H}_i\boldsymbol{x}^b)\right)$. Now let $D = |\boldsymbol{A}|^{1/2} = |\boldsymbol{A}^{-1}|^{-1/2}$. From here simply substitute $\boldsymbol{A}^{-1} = \boldsymbol{P}^{-1} + \boldsymbol{H}_i^T\boldsymbol{R}^{-1}\boldsymbol{H}_i$ into $D$ to obtain

$D = |\boldsymbol{A}^{-1}|^{-1/2}$
$= |\boldsymbol{P}^{-1} + \boldsymbol{H}_i^T\boldsymbol{R}^{-1}\boldsymbol{H}_i|^{-1/2}$
$= |\boldsymbol{P}^{-1}|^{-1/2}|\boldsymbol{I} + \boldsymbol{P}\boldsymbol{H}_i^T\boldsymbol{R}^{-1}\boldsymbol{H}_i|^{-1/2}$
$= |\boldsymbol{P}^{-1}|^{-1/2}|\boldsymbol{I} + \boldsymbol{R}^{-1}\boldsymbol{H}_i\boldsymbol{P}\boldsymbol{H}_i^T|^{-1/2}$  (Sylvester's Determinant Theorem)
$= |\boldsymbol{P}^{-1} + \boldsymbol{P}^{-1}\boldsymbol{R}^{-1}\boldsymbol{H}_i\boldsymbol{P}\boldsymbol{H}_i^T|^{-1/2}$
$= |\boldsymbol{P}^{-1}|^{-1/2}|\boldsymbol{R}^{-1}|^{-1/2}|\boldsymbol{R} + \boldsymbol{H}_i\boldsymbol{P}\boldsymbol{H}_i^T|^{-1/2}$
$\propto |\boldsymbol{R} + \boldsymbol{H}_i\boldsymbol{P}\boldsymbol{H}_i^T|^{-1/2}$ as required. However, as the author stated I also think this proof should be readily available in some textbook as it is just the marginal log-likelihood of a Gaussian density.

---

## Author Comment (AC1) · 4 Jul 2017

article times natbib

[Figure]

**Response to Referees' Comments**

Peter Rayner

July 4, 2017

I thank Amy Braverman and an anonymous referee for their comments. Both have highlighted a series of problems of presentation which I have addressed in the revision. The reviews have also prompted me to read more widely in the statistical literature and realise that the paper is, as I suspected, an application of existing theory. I now point to this theory, and focus more on developing the example and some of the possibilities and problems that arise in biogeochemical applications. I have made a series of general and specific responses to reviewers' comments. I detail the general responses first and address specific concerns from each referee below. I have placed referee comments in Typewriter font and my responses in Roman.

**General Comments**

1. I have removed the appendix and replaced it with a refernce to (*MacKay*, 2003, Ch. 28). thanks to the anonymous reviewer for pointing this out.

2. I have expanded the description of the TRANSCOM inversion as requested by Amy Braverman.

3. I have made the conditioning on the data explicit in the various PDFs.

4. I have made a careful pass through the manuscript to regularise the typography.

**Amy Braverman**

General comments: In general I like this paper a lot. However, I find it extremely difficult to follow because of some type-o's and much notation with which I am not familiar. The notation seems inconsistent in distinguishing between fixed quantities and random ones, and indicating where conditioning has taken place. It is easy at this point to get side-tracked into a discussion of the interpretation of probability. I think that Dr. Braverman's real concern is the discussion of the TRANSCOM case where the target variables and data should be more clear. See general response 2. I agree that it should be clear when conditioning has taken place, see general response 3.

Page 3, lines 6 and 7: Please define the random variables x and Hi . In what sense is P (x|Hi ) "the conventional data assimilation problem"? I hav added some explanatory text to clarify this.

 2. Page 3, line 9: To what "linear model" are you referring? A linear transport model represented by Hi ? What do you mean by "over enough of the relevant pdf's"? Or, do you mean "over enough of the support of the random variable Hi ? I should have said a linear observation operator. I have corrected this and expanded the text.

 3. Page 3, line 12 and 13: Is Hi the same as Hi ? H1 , . . . , HN are defined here as Jacobian matrices corresponding to N different transport models "...with un- knowns defined by

the multivariate Gaussian G(xb , B)...". Which unknowns? I am following the notation of *Rayner et al.* (2016) so that $\mathbf{H}_i$ is the linearised form of $H_i$. This distinction is unnecessary here so I have changed to the linearised form throughout. $\mathbf{x}$ are the continuous variables described in the text now added at the head of the section.

4. Page 3, line 15: "For each Hi our problem is the simple linear Gaussian inver- sion..." What does this mean? What is it you are trying to solve for or infer? Is it the flux that gave rise to the observed concentrations? The problem is more general than fluxes and concentrations although that is a common example and one I use later. Again, I hope the explanatory text at the head of the section explains the meaning of the symbols.

5. Page 3, line 16: "Most importantly for us P (xa |Hi ) is Gaussian." Please define xa . Should it be xa ? These should be bold throughout following *Rayner et al.* (2016). I have corrected this. 4$\mathbf{x}^a$ is the analysis or posterior.

6. Page 3, lines 16 and 17: P (x, Hi ) appears to be a joint distribution of two quanti ties: the vector-valued x and the matrix-valued Hi . It's unclear from the notation whether Hi is a random matrix or a fixed matrix. (On line 21, Hi is treated as random.) My guess is that it is fixed since the right side of the equal sign ap- pears to show the pdf of just one variable; presumably x. Is $\mu$i a vector or a scalar? Please define $\mu$i , Ui , and Wi . I have added these definitions. I have also switched from using $\mathbf{H}_i$ as the variable in the PDF to $i$ since it is the index into the set of observation operators which is the target variable.

The expression P (x, Hi ) = Wi G($\mu$i , Ui ) does not define

a proper pdf unless Wi = 1 since the area under the pdf must
equal one.  A more precise definition of a mixture would be in
terms of ran- P 1 with probability wi , PK dom variables:  X =
K i=1 Ai Xi , Ai = , i=1 wi = 0 otherwise 1, and Xi âĹij G($\mu$i ,
Ui ). I don't agree with this. The expression represents the probability that $i$ is the correct model and $\mathbf{x}$ the value of the continuous target variable. The normalisation requirement is defined by the integral over continuous target variables and sum over models. That is expressed by the extra constraint in the next equation.

7.  Page 3, line 23:  Either x should be bold, or not.  Do
not mix within the same equation.  Also, the notation G($\mu$i ,
Ui )(x) seems is very confusing (to me, at least).  Do you mean
that x is an argument to the function G? Why not write G(x|$\mu$i
, Ui )? See general point above for typography.  The other reviewer also noted the number of arguments in the definition of the Gaussian.  I have regularised this, choosing to make the argument of the function, as well as its parameters, explicit. Note that this is different from the normal shorthand.

8.  Page 3, line 26:  In this equation Hi is treated as a
non-random quantity.  Above in line 21 it was random.  Have you
conditioned on it?  If so, this distribution should be written
as a conditional distribution.  If not, then Wi is a random
variable, not a fixed weight.  $W_i$ represents the probability that the ith model is the true model. I have added an explanation of this at the beginning of the section, hopefully clarifying several of Dr. Braverman's concerns at once.

9.  Page 3, line 26:  I can't check this equation because I
can't follow the derivation in Appendix A. See below.  It appears from reviewer 2's comments that the derivation is a standard result which I now quote.

10.  Page 3, line 27:  I think there is an extra "v" at the

end of this line. removed.

11.  Page 4, line 2:  I assume that xb has a prior distribution somewhere because it is being treated as both random and fixed in various places.  What is the prior distribution? $x^b$ is the mean of the prior distribution for the target variable $x$. The confusion here raises a general question on which I seek editorial guidance. I have relied heavily on the notation and explanations in *Rayner et al.* (2016) thus making the current paper less self-contained.  Should I move away from that and define the notation locally?

12.  Page 4, line 16:  Type-o. I am not seeing this.

13.  Page 4, Footnote is missing. Removed.

Appendix A, through page 12
1.  Page 12, line 18:  K was defined in the main text as a normalizing constant.  What is K(Hi ) here?  Do you mean P (Hi )?
2.  Page 12, line 21:  I am confused by this equation.  G is a function that has an argument and parameters.  What are the parameters and what are the arguments in this expression?  The definitions from Section 2, lines 13 and 14 should be restated here and clarified as indicated earlier.
3.  Page 12, line 23:  Please define $\sigma$.  Why is x in bold while dx is not?
4.  Page 12, line 25:  I find the use of Hi as both a Jacobian and an indicator of model identity to be very confusing.  Why not let Hi be the Jacobian of model i, and introduce a model indicator, say $\Delta$, an integer-value random variable taking values in 1, 2, .  .  .  , M , where M is the number of models?

See general comment 1.

**Reviewer 2**

It is important to note that all the theory from Section 2
is conditional on the same data y being used in all of the
inversions.  Is this the case in the Gurney study?  What can
be done when y differs across models?  Yes, the TRANSCOM inversion
studies kept the PDFs for prior and data constant and changed only the observation
operator. I could imagine weighting model 1 by its match to data $\mathbf{Y}_1$ and model 2 to
data $\mathbf{Y}_2$ but this would get quite tricky if the dimensions were different and might be
difficult to interpret in any case.

 I disagree with the introduction of JIC. Isn't this just the
marginal log-likelihood of the ith component?  I think it
should be described as such.  Also it is well-known that the
marginal log-likelihood penalises for complex models and the
marginal likelihood is commonly employed in model selection.   I
agree and thank the reviewer for sending me back to the literature. I have expanded
the introductory section and reduced the theoretical development accordingly and now
use the conventional terminology. I am more concerned with model weighting than
model selection here but agree that the marginal likelihood rather than its logarithm is
the more useful quantity. The logarithm is still a more convenient tool for presentation
however and I continue to use it for figures.

 Are you sure that if we replace H i BH Ti + R with I we
retrieve the BIC? The BIC plugs-in maximum likelihood estimates
of x into the log-likelihood, while the marginal likelihood
(which you label JIC) integrates out x, which is different.   I

think they are equivalent. The same thing happens with the conventional $\chi^2$ for an inversion. this uses the MAP for the target variables but, in the linear Gaussian case, when one substitutes this value, one obtains an expression involving the innovations and uncertainty projected into observation space.

P3 L30: The comment 'We don't believe that the relative quality of two model depends on the amount of data used to compare then...' is slightly out of place. There are many texts that show that asmyptotically these criteria hold (under some assumptions). If there is reason to believe that the assumptions are not valid (and the criteria are hence not valid) for the flux inversion problem discussed, then some other theoretical foundation for the use of a different criterion is needed. My point is pedagogical rather than theoretical. Modellers tend to use comparison with data as a measure of relative quality. This is quite reasonable but the relative quality of models doesn't diverge as we use more data to compare them. I have rewritten this point.

Related to the previous comment, the theory shows that the posterior flux is a weighted Gaussian mixture with weights Wi . What is the theoretical justifica- tion of using the marginal log-likelihood to weight when combining (as I think is implied when using the expression 'JIC-weighted')? The Wi being degenerate is unfortunate but not a valid reason. The same comment holds for the cross- validation metrics. This was an error of writing rather than calculation. I used the marginal likelihood throughout for weights but was incorrect in several places in calling this the JIC.

P8 L25: I cannot find a reasonable justification as to why one should add R and R sample . There are justifiable alternatives – for example in spatial analysis one would fit a space-time

model to the residuals with nugget and use this for the new
R. This will be guaranteed to be non-singluar and positive
definite.  I think that adding these two matrices can lead to
unforeseen consequences in other settings. I have now expanded this
section and discuss in more detail the task of describing the model contribution to
uncertainty. The reviewer's example is an interesting example. This kind of fitting of
spatial models is certainly desirable if we have enough data to do it. In biogeochemical
applications this is rare and we need something else. I agree that the addition of the
sample covariance of residuals is ad hoc and I now describe it as an example, along
with another of the sample covariance of a priori simulations.

Other Comments

 P1 L24:  I do not agree with the comment 'When structural
uncertainty enters the problem only ensemble methods are
available'.  Although less direct, cannot one introduce an
additional unstructured residual term in the model and see if
that dominates? I agree, the comment was too strong and I have moderated it.

Section 2 and Appendix A need work to make the notation
consistent.  Just some things I picked up:  – Hi should be
bold everywhere – The lack of conditioning on the data y in
all equations makes it hard to distinguish between prior and
posterior distributions.  Also the use of K(H i ) as prior
is confusing since K is a normalising constant in Equation
(6).  – The matrix B in Section 2 has become P in Appendix A.
– In Appendix A, the PDF G(Âůˇ) takes three arguments, while in
Section 2 it only takes 2.  – Appendix A needs to be cleaned
up, there are expressions like 'mathbf' appearing, $\mu$ is not
bold, the subscript i could be apparent on the LHS but not on
the RHS, etc.  See also my comment on clarifying the maths in

[Figure]

Section A further below.

P12: I found that the maths from Equation (A10) onwards
becomes a bit obscure. The result is correct, however I think
more details are needed. (some algebra deleted) However,
as the author stated I also think this proof should be
readily available in some textbook as it is just the marginal
log-likelihood of a Gaussian density. I group all these comments to-
gether since I have dealt with them all by essentially quoting the standard result on
marginal likelihood and leaving out most of the algebra.

P3 L27: Instead of "assumption that we must choose one" I
would instead say "assumption that the sum of the probabilities
of the models given the data equals one." Strictly speaking,
unless you are using a Bayesian estimator you do not need to
choose any specific one model. I was trying to make a different point that
"none of the above" is not an option. I have reworded this.

P4 L7: The statement 'Eq.6 is also the same expression as the
maximum likeli- hood estimate in Michalak et al. (2005)' is
inaccurate. Estimate of what? Eq.6 is the marginal likelihood
of the data under the ith component. But the expressions *are* the same. I
now think this is because they represent the same thing, the marginal likelihood for a
hyperparameter. I mention this in passing now but want to keep the paper focused on
the practical use of the machinery.

P5 L32: What is the difference between JIC/N and JIC if N
is constant? Does scaling affect any of the results and
conclusions? Not here since we only use one dataset. See the above response on
relative model quality.

P7 L9: Why 'model seven'? From Figure 1 it looks like Model 8

`is the best model?` Apologies, this was a numbering from 0 vs numbering from 1 problem, I have corrected the text.

**References**

MacKay, D. J. C., *Information Theory, Inference, and Learning Algorithms*, Cambridge University Press, available from `http://www.inference.phy.cam.ac.uk/mackay/itila/`, 2003.

Rayner, P., A. M. Michalak, and F. Chevallier, Fundamentals of data assimilation, *Geoscientific Model Development Discussions*, *2016*, 1–21, doi:10.5194/gmd-2016-148, 2016.

---

## Author Response (AR1)

**Response to Referees' Comments**

Peter Rayner

October 11, 2018

I thank Amy Braverman and an anonymous referee for their comments. Both have highlighted a series of problems of presentation which I have addressed in the revision. The reviews have also prompted me to read more widely in the statistical literature and realise that the paper is, as I suspected, an application of existing theory. I now point to this theory, and focus more on developing the example and some of the possibilities and problems that arise in biogeochemical applications. I have made a series of general and specific responses to reviewers' comments. I detail the general responses first and address specific concerns from each referee below. I have placed referee comments in Typewriter font and my responses in Roman.

**General Comments**

1. I have removed the appendix and replaced it with a refernce to (*MacKay*, 2003, Ch. 28). thanks to the anonymous reviewer for pointing this out.

2. I have expanded the description of the TRANSCOM inversion as requested by Dr. Braverman.

3. I have made the conditioning on the data explicit in the various PDFs.

4. I have made a careful pass through the manuscript to regularise the typography.

**Amy Braverman**

General comments: `In general I like this paper a lot. However, I find it extremely difficult to follow because of some type-os and much notation with which I am not familiar. The notation seems inconsistent in distinguishing between fixed quantities and random ones, and indicating where conditioning has taken place.` It is easy at this point to get side-tracked into a discussion of the interpretation of probability. I think that Dr. Braverman's real concern is the discussion of the TRANSCOM case where the target variables and data should be more clear. See general response 2. I agree that it should be clear when conditioning has taken place, see general response 3.

Page 3, lines 6 and 7:  Please define the random variables x and Hi .  In what sense is P (x|Hi ) the conventional data assimilation problem"? I hav added some explanatory text to clarify this.

2.  Page 3, line 9:  To what linear model" are you referring? A linear transport model represented by Hi ?  What do you mean by over enough of the relevant pdfs"?  Or, do you mean over enough of the support of the random variable Hi ? I should have said a linear observation operator. I have corrected this and expanded the text. "support" is precisely the language I needed.

3.  Page 3, line 12 and 13:  Is Hi the same as Hi ?  H1 , . . . , HN are defined here as Jacobian matrices corresponding to N different transport models ...with un- knowns defined by the multivariate Gaussian G(xb , B)...".  Which unknowns? I am following the notation of *Rayner et al.* (2016) so that $\mathbf{H}_i$ is the linearised form of $H_i$. This distinction is unnecessary here so I have changed to the linearised form throughout. $\mathbf{x}$ are the continuous variables described in the text now added at the head of the section.

4.  Page 3, line 15:  For each Hi our problem is the simple linear Gaussian inver- sion..." What does this mean?  What is it you are trying to solve for or infer?  Is it the flux that gave rise to the observed concentrations? The problem is more general than fluxes and concentrations although that is a common example and one I use later. Again, I hope the explanatory text at the head of the section explains the meaning of the symbols.

5.  Page 3, line 16:  Most importantly for us P (xa |Hi ) is Gaussian." Please define xa .  Should it be xa ? These should be bold throughout following *Rayner et al.* (2016). I have corrected this. $\mathbf{x}^a$ is the analysis or posterior.

6.  Page 3, lines 16 and 17:  P (x, Hi ) appears to be a joint distribution of two quanti ties:  the vector-valued x and the matrix-valued Hi .  Its unclear from the notation whether Hi is a random matrix or a fixed matrix.  (On line 21, Hi is treated as random.)  My guess is that it is fixed since the right side of the equal sign ap- pears to show the pdf of just one variable; presumably x.  Is i a vector or a scalar?  Please define i , Ui , and Wi . I have added these definitions. I have also switched from using $\mathbf{H}_i$ as the variable in the PDF to $i$ since it is the index into the set of observation operators which is the target variable.

The expression P (x, Hi ) = Wi G(i , Ui ) does not define a proper pdf unless Wi = 1 since the area under the pdf must equal one.  A more precise definition of a mixture would be in terms of ran- P 1 with probability wi , PK dom variables: X = K i=1 Ai Xi , Ai = , i=1 wi = 0 otherwise 1, and Xi  G(i , Ui ). I don't agree with this. The expression represents the probability that $i$ is the correct model and $\mathbf{x}$ the value of the continuous target variable. The normalisation requirement is defined by the integral over continuous target variables and sum over

models. That is expressed by the extra constraint in the next equation. I have made this more explicit by writing $W_i$ as $P(H_i$.

```
    7.  Page 3, line 23:  Either x should be bold, or not.
Do not mix within the same equation.  Also, the notation G(i
, Ui )(x) seems is very confusing (to me, at least).  Do you
mean that x is an argument to the function G? Why not write
G(x|i , Ui )?
```
Done throughout and a definition has been added in Section 2.

```
    8.  Page 3, line 26:  In this equation Hi is treated as
a non-random quantity.  Above in line 21 it was random.  Have
you conditioned on it?  If so, this distribution should be
written as a conditional distribution.  If not, then Wi is
a random variable, not a fixed weight.
```
I have rewritten the equations to make the probabilities explicit and expanded the explanation at the start of Section 2.

```
    9.  Page 3, line 26:  I cant check this equation because
I cant follow the derivation in Appendix A. See below.
```
It appears from reviewer 2's comments that the derivation is a standard result which I now quote.

```
    10.  Page 3, line 27:  I think there is an extra v" at
the end of this line.
```
removed.

```
    11.  Page 4, line 2:  I assume that xb has a prior distribution
somewhere because it is being treated as both random and fixed
in various places.  What is the prior distribution?
```
$\mathbf{x}^b$ is the mean of the prior distribution for the target variable $\mathbf{x}$. The confusion here raises a general question on which I seek editorial guidance. I have relied heavily on the notation and explanations in *Rayner et al.* (2016) thus making the current paper less self-contained. Should I move away from that and define the notation locally?

```
    12.  Page 4, line 16:  Type-o.
```
Corrected.

```
    13.  Page 4, Footnote is missing.
```
Removed.

```
    Appendix A, through page 12
1.  Page 12, line 18:  K was defined in the main text as a
normalizing constant.  What is K(Hi ) here?  Do you mean P
(Hi )?
2.  Page 12, line 21:  I am confused by this equation.  G is
a function that has an argument and parameters.  What are the
parameters and what are the arguments in this expression?  The
definitions from Section 2, lines 13 and 14 should be restated
here and clarified as indicated earlier.
3.  Page 12, line 23:  Please define . Why is x in bold while
dx is not?
4.  Page 12, line 25:  I find the use of Hi as both a Jacobian
and an indicator of model identity to be very confusing.  Why
not let Hi be the Jacobian of model i, and introduce a model
indicator, say , an integer-value random variable taking values
in 1, 2, .  .  .  , M , where M is the number of models?
```
See general comment 1.

**Reviewer 2**

It is important to note that all the theory from Section 2
is conditional on the same data y being used in all of the
inversions.  Is this the case in the Gurney study?  What can
be done when y differs across models? Yes, the TRANSCOM inversion studies kept the PDFs for prior and data constant and changed only the observation operator. This is now noted at the start of Section 3. the formalism as expressed here would struggle with different dimensions for different **y** and this is now noted at the end of Section 6.

 I disagree with the introduction of JIC. Isnt this just
the marginal log-likelihood of the ith component?  I think
it should be described as such.  Also it is well-known that
the marginal log-likelihood penalises for complex models and
the marginal likelihood is commonly employed in model selection.
I agree and thank the reviewer for sending me back to the literature. I have expanded the introductory section and reduced the theoretical development accordingly and now use the conventional terminology. I am more concerned with model weighting than model selection here but agree that the marginal likelihood rather than its logarithm is the more useful quantity. The logarithm is still a more convenient tool for presentation however and I continue to use it for figures.

 Are you sure that if we replace H i BH Ti + R with I we
retrieve the BIC? The BIC plugs-in maximum likelihood estimates
of x into the log-likelihood, while the marginal likelihood
(which you label JIC) integrates out x, which is different.
I think they are equivalent. The same thing happens with the conventional $\chi^2$ for an inversion. this uses the MAP for the target variables but, in the linear Gaussian case, when one substitutes this value, one obtains an expression involving the innovations and uncertainty projected into observation space.

 P3 L30:  The comment We dont believe that the relative quality
of two model depends on the amount of data used to compare
then... is slightly out of place.  There are many texts that
show that asmyptotically these criteria hold (under some assumptions).
If there is reason to believe that the assumptions are not
valid (and the criteria are hence not valid) for the flux inversion
problem discussed, then some other theoretical foundation for
the use of a different criterion is needed. My point is pedagogical rather than theoretical. Modellers tend to use comparison with data as a measure of relative quality. This is quite reasonable but the relative quality of models doesn't diverge as we use more data to compare them. Recall also that this paper is primarily concerned with model weighting rather than selection. I regard it as pathological that the weighting collapses towards a single model as the amount of data increases. I have rewritten this point.

 Related to the previous comment, the theory shows that
the posterior flux is a weighted Gaussian mixture with weights
Wi .  What is the theoretical justifica- tion of using the

marginal log-likelihood to weight when combining (as I think
is implied when using the expression JIC-weighted)? The Wi
being degenerate is unfortunate but not a valid reason.  The
same comment holds for the cross- validation metrics. the Gaussian mixture is a result rather than an assumption. Part of the confusion was caused by
an error of writing rather than calculation. I used the marginal likelihood throughout
for weights but was incorrect in several places in calling this the JIC.

    P8 L25:  I cannot find a reasonable justification as to
why one should add R and R sample .  There are justifiable
alternatives  for example in spatial analysis one would fit
a space-time model to the residuals with nugget and use this
for the new R. This will be guaranteed to be non-singluar and
positive definite.  I think that adding these two matrices
can lead to unforeseen consequences in other settings. I have
now expanded this section and discuss in more detail the task of describing the model
contribution to uncertainty.  The reviewer's example is another interesting approach.
This kind of fitting of spatial models is certainly desirable if we have enough data to
do it.  In this case we don't but the problem still needs a resolution albeit it mperfect.  I agree that the addition of the sample covariance of residuals is ad hoc and I
now describe it as an example, along with another of the sample covariance of a priori
simulations.

    Other Comments

    P1 L24:  I do not agree with the comment When structural
uncertainty enters the problem only ensemble methods are available.
Although less direct, cannot one introduce an additional unstructured
residual term in the model and see if that dominates? I agree,
the comment was too strong and I have moderated it.

    Section 2 and Appendix A need work to make the notation
consistent.  Just some things I picked up:  Hi should be bold
everywhere  The lack of conditioning on the data y in all equations
makes it hard to distinguish between prior and posterior distributions.
Also the use of K(H i ) as prior is confusing since K is a
normalising constant in Equation (6).  The matrix B in Section
2 has become P in Appendix A.  In Appendix A, the PDF G() takes
three arguments, while in Section 2 it only takes 2.  Appendix
A needs to be cleaned up, there are expressions like mathbf
appearing,  is not bold, the subscript i could be apparent
on the LHS but not on the RHS, etc.  See also my comment on
clarifying the maths in Section A further below.
P12:  I found that the maths from Equation (A10) onwards becomes
a bit obscure.  The result is correct, however I think more
details are needed.  (some algebra deleted) However, as the
author stated I also think this proof should be readily available
in some textbook as it is just the marginal log-likelihood
of a Gaussian density. I group all these comments together since I have
dealt with them all by essentially quoting the standard result on marginal likelihood

and leaving out most of the algebra.

    `P3 L27:  Instead of assumption that we must choose one I would instead say assumption that the sum of the probabilities of the models given the data equals one. Strictly speaking, unless you are using a Bayesian estimator you do not need to choose any specific one model.` I think here we *are* using a Bayesian estimator. I was trying to make a different point that "none of the above" is not an option. I have reworded this.

    `P4 L7:  The statement Eq.6 is also the same expression as the maximum likeli- hood estimate in Michalak et al. (2005) is inaccurate.  Estimate of what?  Eq.6 is the marginal likelihood of the data under the ith component.` But the expressions *are* the same. I now think this is because they represent the same thing, the marginal likelihood for a hyperparameter. I mention this in passing now but want to keep the paper focused on the practical use of the machinery.

    `P5 L32:  What is the difference between JIC/N and JIC if N is constant?  Does scaling affect any of the results and conclusions?` Not here since we only use one dataset. See the above response on relative model quality.

    `P7 L9:  Why model seven? From Figure 1 it looks like Model 8 is the best model?` Apologies, this was a numbering from 0 vs numbering from 1 problem, I have corrected the text.

**References**

[revised manuscript text omitted]

---

## Referee Report (RR1)

**Round 2 Referee Report for**
**Data Assimilation using an Ensemble of Models:**
**A hierarchical approach**

**General comments:**

I read the new version of the paper from the top, and have a new set of specific comments and questions that are given below. I am still stuck on Section 2. However, we have made substantial progress because the increased clarity of the new version makes it now possible to identify the ambiguities that are hanging me up.

First, though, I have an emphatic comment: I *do* think the paper should be self-contained. I am referring here to the author's response to my item 11 in the first round of review, in which he states, "I have relied heavily on the notation and explanations in Rayner et. al. (2016) thus making the current paper less self-contained. Should I move away from that and define the notation locally?" My opinion is a definite "yes". Without those definitions and explanations, it's quite difficult to follow what's going on. Unless you want this paper to appeal only to those who are quite familiar with previous work, I feel it's essential that this paper explain all the notational conventions that are used, and provide adequate background.

**Specific comments:**

1. Page 2, lines 24 to 28: What does "This required running optimized fluxes through the forward model used to generate the Jacobians" have to do with challenging equal weighting? What is TM3?

2. Page 3, Equation (1): Here is a case where you have referenced Rayner et. al. (2016). I find the material here impossible to understand without going back to that 2016 article, and even so, it's not clear where this formula is coming from. In Rayner et. al. (2016), a similar version of current paper's Equation (1) appears as Equation (2), but it doesn't look right to me: is it missing and integral? For the left-hand side to be $p(x)$, you would have to integrate out all the other variables. You appear to be headed that way in Equation (1) of the current paper by integrating out $\mathbf{y}^t$, as if $H(\cdot)$ is a deterministic function of $\mathbf{x}$. However, this is never stated, and is contrary to both the notation and treatment of $H$ later.

3. It's difficult to tell what the derivations in the first part of Section 2 are trying to show. It looks to me that you want to end up with Equation (3), which is an expression for

$P(\mathbf{x}, \mathbf{H}_i|\mathbf{y})$, but this is only an intermediate step towards getting $p(\mathbf{x}|\mathbf{y})$:

$$p(\mathbf{x}|\mathbf{y}) = \frac{\sum_i p(\mathbf{x}, \mathbf{H}_i, \mathbf{y})}{p(\mathbf{y})} = \frac{\sum_i p(\mathbf{x}|\mathbf{H}_i, \mathbf{y})\, p(\mathbf{H}_i, \mathbf{y})}{p(\mathbf{y})},$$

$$= \frac{\sum_i p(\mathbf{x}|\mathbf{H}_i, \mathbf{y})\, p(\mathbf{H}_i|\mathbf{y})\, p(\mathbf{y})}{p(\mathbf{y})} = \sum_i p(\mathbf{x}|\mathbf{H}_i, \mathbf{y})\, p(\mathbf{H}_i|\mathbf{y}). \qquad (1)$$

Moreover, you want the expression for $p(\mathbf{x}|\mathbf{y})$ to factor in such a way that it involves estimating $p(\mathbf{H}_i|\mathbf{y})$ because those are the weights for the transport models, and you are interested in those for their own sakes. I think this argument could be made more clearly if you started Section 2 by stating that the ultimate goal is to obtain the moments of $p(\mathbf{x}|\mathbf{y})$, which can be factored in different ways, and the particular factorization above is the most informative because it involves estimating the weights $p(\mathbf{H}_i|\mathbf{y})$.

4. In the footnote on page 3 you explain that $\mathbf{H}_i$ is intended to be an indicator variable that really represents the index into a set of transport models. You also say that $\mathbf{H}_1, \ldots, \mathbf{H}_N$ are the Jacobians of those transport models (line 26). Elsewhere, $H_i$ is not bold (Equation (2)). These conventions should all be described in the main text (no footnote) and the meaning of bold versus non-bold should be clarified. I suspect your use of non-bold $H_i$ and non-bold $x$ in Equation (2) is because you are stating a generic result, and you are not specifically referring to $\mathbf{H}_i$ and $\mathbf{x}$ used the rest of the text in this section. Please explain that.

5. Line 27, page 3: Please define $\mathbf{y}^O$. I get that it is the mean of the random vector that represents the observations, but is it different that $\mathbf{y}^t$? It probably could be, but are you making any assumptions about that? Also, here you treat $\mathbf{x}^b$ as the mean of the Gaussian distribution of the random variable $\mathbf{x}$, but Equation (1) treats it like a random variable ($p(\mathbf{x}|\mathbf{x}^b)$). Of course, it is possible that it could be both if the model was hierarchical and specified a prior distribution on $\mathbf{x}^b$, but if that's the case it should be stated. I suspect that this is really just notation given that you write, $G(\mathbf{x}|\boldsymbol{\mu}, \mathbf{C})$ on line 28 (if $\mathbf{C}$ is bold, then $\boldsymbol{\mu}$ should also be bold). Finally, the expression "uncertainty covariance" is somewhat confusing, at least to me: should it just be "covariance"?

6. Lines 29–30 on page 3 and Equation (3): I don't understand why this is here, but perhaps that is because my understanding of what you are trying to do relies on expressions I wrote above for item 3 (my Equation (1)). The final expression for $p(\mathbf{x}|\mathbf{y})$ there is already in terms of $p(\mathbf{x}|\mathbf{H}_i, \mathbf{y})$. You then write, "Thus our posterior for the ensemble is a mixture of Gaussians...", which I agree with. We both have $p(\mathbf{H}_i|\mathbf{y})$ (I note that you have now switched to using capital "$P$" for probability instead of "$p$" used earlier– it's a minor thing, but it would be better to be consistent), and the remaining term I call $p(\mathbf{x}|\mathbf{H}_i, \mathbf{y})$ and you call $G(\mathbf{x}|\mathbf{x}_i^a, \mathbf{A}_i)$. It might be helpful to clarify this correspondence in the text since it ties back to the ultimate objective of expressing the posterior $p(\mathbf{x}|\mathbf{y})$ in a special way that admits the mixture of Gaussians representation.

7. Lines 2–3, page 4: When you say, "As usual with a joint PDF we obtain the marginal probability for a variable by integrating over all others", to what are you referring? Are you justifying Equation (4)?

8. Equation (4): There are a few things about this that need to be addressed or explained. First, you stated earlier in the footnote on page 3 that $\mathbf{H}_i$ is a stand-in for an index random variable that distinguishes between transport models, but you use $\mathbf{H}_i$ anyway to remind the reader to what this index refers. If that remains true, then $\mathbf{H}_i$ is a discrete variable here, not a continuous one. If that's the case, then $P(\mathbf{H}_i)$ is not Gaussian, and I don't think the right-hand-side of the equation makes sense. In Michalak et. al. (2005), the target of inference is $\boldsymbol{\theta}$ which is a vector of continuously-valued variance parameters, so it makes sense there. I think what you are trying to do with this expression is to obtain the set weights associated with models represented by $\mathbf{H}_i$ as in Raftery et. al., (2005) which you cite. Alternatively, maybe you have changed the notation implicitly to treat $\mathbf{H}_i$ (or more properly vec($\mathbf{H}_i$)) as a Gaussian random vector. If so, please explain.

9. Lines 7–10, page 4: Several issues here. First, the words of the first sentence in Section 2.1 provide an example of where $\mathbf{x}^b$ is now discussed as if it were a random variable rather a parameter (in contrast to its use earlier in the paper). Is $\mathbf{x}^b$ a parameter of the prior distribution of $\mathbf{x}$ or is it a random draw from that distribution? Only in the latter case does the notion of independence from $\mathbf{y}$ make sense. Second, Equation (4) as stated is not the probability of simulating the observations ($\mathbf{y}$); it is the probability of $\mathbf{H}_i$ given the observations. Should it be $p(\mathbf{y}|\mathbf{H}_i, \mathbf{x})$? Third, I question assertion made in Michalak et. al. (2005), Section 6.4, Equation (4) that Equation (2) of that paper can be written,

$$p(\mathbf{x}) \propto G(\mathbf{x} - \mathbf{x}^b, \mathbf{B}) \times G(\mathbf{H}(\mathbf{x}) - \mathbf{y}, \mathbf{R}).$$

Equation (2) in Michalak et. al. (2005) is $p(x) \propto p(x|x^b) \times p(y^t|y) \times p(y^t|H(x))$. It appears to me that $p(x|x^b)$ (or $p(\mathbf{x}|\mathbf{x}^b)$ using the notation of the paper under review) is missing from the expression above. Finally, also $G(\mathbf{x} - \mathbf{x}^b, \mathbf{B})$ is ambiguous at best and nonsense at worst: do they mean $G(\mathbf{x} - \mathbf{x}^b|\mathbf{x}^b, \mathbf{B})$ and $G(\mathbf{H}(\mathbf{x}) - \mathbf{y}|\mathbf{H}(\mathbf{x}), \mathbf{R})$?

10. Lines 17–19, page 4: $P(\mathbf{H}_i)$ appears several times in this passage. Do you mean $P(\mathbf{H}_i|\mathbf{y})$?

11. Lines 22–28, page 4: What's the point of this second-to-last paragraph of Section 2.1? Is it simply to draw a line between the more familiar concept of $\chi^2$ in the literature and the work here? You do use it in the next paragraph (and in Section 2.2), so perhaps these should all be combined into one paragraph? That would make it clear why $\chi^2$ is being defined. Also, I don't understand the calculation given in lines 25–27.

12. Section 2.2: The statement that neither AIC nor BIC "take account of different prior uncertainties among parameters or different sensitivities of the observations to these parameters" is mysterious to me. That is certainly true, but that's not their purpose. Since I am confused about what $\mathbf{H}_i$ means here notationally, and that makes it hard to understand what you are driving at.

---

## Referee Report (RR2)

**Referee's Report: "Data Assimilation Using an Ensemble of Models: A Hierarchical Approach" by Peter Rayner, submitted to *Atmospheric Chemistry and Physics*.**

This paper considers the structural uncertainty for the inversion problem and applies the Bayesian model averaging methodology (e.g., Raftery et al., 1997; Hoeting et al., 1999) to obtain the posterior ensemble mean and variance of $\mathbf{x}$ for a number of inversion models. Compared with simply assigning equal weights to each inversion model, the method in this paper is statistically more appropriate for obtaining the posterior statistics of the ensemble.

**1 Major Comments**

1. Some notations in this paper are not very consistent. For example, at the beginning of Section 2, the author used $p(\cdot)$ for probability density function (PDF), but later on $P(\cdot)$ was used for PDF. In addition, for function $H_i(\cdot)$ and matrix $\mathbf{H}_i$, it is better to add some notes to make a clear distinction. Last, the criterion L in Equation (5) is not italic, but later on it appears in italic font and hence can be a bit confusing.

2. The conditional densities in Equations (1) and (4) are also conditional on $\mathbf{x}^b$, and hence the author should mention $\mathbf{x}^b$ is omitted for notation simplicity. Besides, is the prior mean $\mathbf{x}^b$ treated as a fixed or random quantity in this paper?

3. Page 4, Line 7, the author mentioned that "Provided $\mathbf{x}^b$ and $\mathbf{y}$ are independent, $\mathbf{R} + \mathbf{H}_i\mathbf{B}\mathbf{H}_i^T$ is the variance of the prior mismatch $\mathbf{y} - \mathbf{H}_i\mathbf{x}^b$...", which seems to be inappropriate. This is because the matrix $\mathbf{B}$ is the covariance matrix of $\mathbf{x}$, not of the prior mean $\mathbf{x}^b$.

4. Page 6, in Figure 1, why the weight of model 3 is so small for the tuned case, compared with other two cases?

5. The author claims that Equation (7) is the variance of the ensemble, which seems to be incorrect. From the formulation, it seems to be the mean squared (prediction) error for $\mathbf{x}$.

6. Page 7, Figure 2: The titles of boxplots are repeated for each row but it is supposed that the results for all the 22 regions are reported. The author should double check whether this figure is correctly produced.

7. For Equations (8) and (9), it is better to give the mathematical definition of the mean terms (e.g., the mean of $\mathbf{H}(x)_i^b$); also the superscript $a$ is missed in Equation (9). Could the author provide more motivations for using $\mathbf{R}_{i,j}^{\text{prior}}$ and $\mathbf{R}_{i,j}^{\text{sample}}$?

8. Page 9, Line 6: The author pointed out that the residual covariances have the largest values for a few terrestrially-influenced stations such as Baltic Sea and so on. A figure showing the residual covariances can be added to support this claim.

9. Page 10, for the section of computational aspects: Provided that $\mathbf{R}$ is a sparse matrix (e.g., diagonal), I think the computational trick is to use a low-rank matrix to approximate $\mathbf{H}_i\mathbf{B}\mathbf{H}_i^T$; then we can resort to the Sherman-Woodbury-Morrison inversion formula to compute the inverse of $(\mathbf{H}_i\mathbf{B}\mathbf{H}_i^T + \mathbf{R})$ and the Sylvester's theorem to compute its determinant (e.g, Cressie and Johannesson, 2008; Sang and Huang, 2012). The author may add a bit more details to make the computational strategy more clear.

10. Page 12, Figure 4: Similar to Figure 2, the results seem to be repeated and not all the regions' statistics are reported. The author should double check whether the figure is correctly produced.

**2 Minor Comments**

1. Page 1, Line 23, the right bracket should be removed.

2. Page 2, Line 12, "discreet" should be "discrete."

3. Page 3, Line 2: "the" in "the standard data assimilation..." should be capitalized. Similarly, Page 6, Line 11: "the" in "the variance is calculated as" should be capitalized. The author needs to double check whether there are similar typos in the paper.

4. The author refers the Equation (1) but I do not see Equation (1) in the context.

5. Page 4, in the second and third paragraph, it seems that $P(\mathbf{H}_i)$ should be $P(\mathbf{H}_i|\mathbf{y})$.

6. Page 4, Line 23: "...$\chi^2$ is equal to the number of observations..." should be "... the expected value of $\chi^2$ is equal to the number of observations..."

7. Page 7, Line 1: "The Superscripts * indicates we consider..." should be "'The superscript * indicates we consider..."

8. Page 9, Line 6: "Eq. 9 and Eq. 8" should be "Eq. 8 and Eq. 9".

9. Page 10: the math symbols, $X^b$ and $X^a$ should be $\mathbf{x}^b$ and $\mathbf{x}^a$, respectively.

10. Page 11: in the caption of Figure 3, the author should give the full name of "JIC."

**References**

Cressie, N. and G. Johannesson (2008). Fixed rank kriging for very large spatial data sets. *Journal of the Royal Statistical Society: Series B (Statistical Methodology) 70*, 209–226.

Hoeting, J. A., D. Madigan, A. E. Raftery, and C. T. Volinsky (1999). Bayesian model averaging: A tutorial. *Statistical Science 14*, 382–401.

Raftery, A. E., D. Madigan, and J. A. Hoeting (1997). Bayesian model averaging for linear regression models. *Journal of the American Statistical Association 92*, 179–191.

Sang, H. and J. Huang (2012). A full scale approximation of covariance functions for large spatial data sets. *Journal of the Royal Statistical Society: Series B (Statistical Methodology) 74*, 111–132.

---

## Referee Report (RR3)

**Referee's Report: "Data Assimilation Using an Ensemble of Models: A Hierarchical Approach" by Peter Rayner, revised for *Atmospheric Chemistry and Physics*.**

I found that the overall presentation of this paper has improved a lot, and most of my comments have been addressed. The author has changed the writings in a few places and some additional comments are as follows:

**1 Major Comments**

1. In Equation (1), the author introduced the notation of observed observations, $\mathbf{y}^o$. I am wondering whether the conditional probability $p(\mathbf{y}|\mathbf{y}^o)$ in the rhs of Equation (1) should be $p(\mathbf{y}^o|\mathbf{y})$, which is the data model when $\mathbf{y}$ denotes the underlying true data process. In the meanwhile, to be clearer, $p(\mathbf{x}, \mathbf{y})$ may be written as $p(\mathbf{x}, \mathbf{y}|\mathbf{x}^b, \mathbf{y}^o)$.

2. Page 3, Line 26: the sentence "Combing Equation 1 and Equation 3" is not necessary, since Equation (4) can be directly obtained based on the chain rule of probabilities.

3. Page 4, Line 16: in Equation (6), the proportional symbol $\propto$ should be $=$, since Equation (4) holds and $p(\mathbf{x}|\mathbf{y}, H_i) = G(\mathbf{x}, \mathbf{x}_i^a, \mathbf{A}_i)$ (Equation (5)).

4. Page 4, Line 20: In Equation (7), $p(\mathbf{H}_i|\mathbf{y})$ should be $p(\mathbf{H}_i|\mathbf{y}^o)$, since $\mathbf{y}^o$ appears in the rhs of Equation (7), and Equation (7) is derived based on $p(\mathbf{H}_i|\mathbf{y}^o) \propto \int p(\mathbf{y}^o|\mathbf{H}_i, \mathbf{x}, \mathbf{R}) \cdot p(\mathbf{H}_i|\mathbf{x}) \cdot p(\mathbf{x}|\mathbf{x}^b, \mathbf{B}) d\mathbf{x}$.

5. For Equation (7), I guess that the constant $K$ should have a subscript $i$, since $K$ depends on the prior distribution of different parameter models, $p(\mathbf{H}_i)$; also it is better to mention that $K$ is a constant in the context.

6 In Figure 2 and Figure 5, I still noticed that the titles of different boxes are **repeated**, and they are not the distinct names for the 22 land-ocean regions of the TRANSCOM inter-comparison. The author needs to double check whether these results are correctly produced.

**2 Minor Comments**

1. Page 1, Line 22: The "PDF" needs to be explained.

2. Page 4, Line 9, the covariance $B$ should have a bold font.

3. Page 4, Line 11, the brackets of the reference "Rayner et al., 2018, Section 6.4" should be removed.

4. Page 5, Line 26: The AIC should be $2M + \chi^2$ with the penalty factor 2.

5. Page 9, Line 29: In Equation (14), the second $(\mathbf{Hx}_j^b - \mu_j^b)$ should have a transpose, $(\mathbf{Hx}_j^b - \mu_j^b)^T$; similarly, in Equation (15), the second $(\mathbf{Hx}_j^a - \mathbf{y}_i)$ should be $(\mathbf{Hx}_j^a - \mathbf{y}_i)^T$.

---

## Referee Report (RR4)

Round 3 Referee Report for
*Data Assimilation using an Ensemble of Models:*
*A hierarchical approach*

**General comments:**

The paper is significantly improved, but there are still some issues and ambiguities with Section 2. I can't fully comment on the remainder of the paper without sorting those out first. That said, the paper reads much better, to me at least, from start to finish.

**Specific comments:**

1. Page 3, lines 10 to 14 and Equation (1): The notation here is (still) ambiguous, and I do not follow how the left and right sides of Equation (1) are equal. You refer to $\mathbf{x}^b$ as the background, which I now understand to be the mean of the prior distribution of $\mathbf{x}$. The prior variance is not specified, so I am not sure how to interpret $p(\mathbf{x}|\mathbf{x}^b)$. You state elsewhere that Gaussian distributions are assumed so the variance would need to be specified. Same issue for $p(\mathbf{y}|\mathbf{y}^o)$ and $p(\mathbf{y}|H(\mathbf{x}))$. You say, "$\mathbf{y}$ represents the observations", but then "$o$ represents the observed value". So what is the difference between $\mathbf{y}$ and $\mathbf{y}^o$? My interpretation of Equation (1) as it is written is,

$$\mathbf{x} \sim G(\mathbf{x}^b, \mathbf{V_x}), \qquad \mathbf{y} \sim G(\mathbf{y}^o, \mathbf{V_{y1}}) \qquad \mathbf{y} \sim G(H(\mathbf{x}), \mathbf{V_{y2}}),$$
$$\mathbf{x} = \mathbf{x}^b + \boldsymbol{\epsilon}, \qquad \mathbf{y} = \mathbf{y}^o + \boldsymbol{\delta}, \qquad \mathbf{y} = H(\mathbf{x}) + \boldsymbol{\eta},$$
$$= H(\mathbf{x}^b + \boldsymbol{\epsilon}) + \boldsymbol{\eta}.$$

This is as far as I can get without clarification on $\mathbf{y}$ and $\mathbf{y}^o$. In any case, it is not obvious to me why one would take the product of these conditional distributions to obtain $p(\mathbf{x}, \mathbf{y})$. I don't think it would be obvious to a general reader either.

2. Page 4, lines 7 to 9. Here the prior covariance matrices of the distributions $p(\mathbf{x}|\mathbf{x}^b)$ and $p(\mathbf{y}|\mathbf{y}^o)$ are finally defined as $\mathbf{B}$ and $\mathbf{R}$, respectively. On line 9, it says "$\mathbf{y}^o$ is the observed value with uncertainty covariance $\mathbf{R}$", but if $\mathbf{y}^o$ is the mean of the prior distribution $\mathbf{y}$, then $\mathbf{R}$ should be the covariance matrix of the prior distribution of $\mathbf{y}$. There is nothing in your model about $\mathbf{y}^o$ being a random variable (this would imply a hierarchical model) up to this point. Did you mean to say, "$\mathbf{y}$ is the observed value with uncertainty covariance $\mathbf{R}$"?

3. Page 4, line 16, Equation 6: shouldn't this be an equality?

4. Page 4, line 17, This is nit-picking at some level, but you write "Thus, $p(\mathbf{x}, \mathbf{H}_i|\mathbf{y})$ is a sum of Gaussian Distributions..." Technically, it's not the distribution that is a sum of Gaussians, it's the random variable that has a distribution that is a Gaussian mixture.

5. Page 4, line 20, Equation 7: Shouldn't the left-hand side of Equation 7 be $p(\mathbf{H}_{i|\mathbf{y}^o})$? There is no term involving $\mathbf{y}$ on the right. Also, I think you are missing a comma at the end of line 19, and a period at the end of Equation 7.

6. Page 4, line 23: You say "Provided $p(\mathbf{x})$ (the prior distribution for $\mathbf{x}$) and $\mathbf{y}$ are independent...". Another nit-pick: it's not the distributions that are independent, it's the random variables. More importantly, how can $\mathbf{x}$ and $\mathbf{y}$ possibly be independent? Isn't $\mathbf{y} = H(\mathbf{x})$ given the last term on the right-hand side of Equation 1? I don't have the Tarantola book handy, so I am not sure what the Jacobian rule of probabilities is (will attempt to look this up). I think what you are trying to do here is to justify adding the variances and ignoring covariance between $\mathbf{x}$ and $\mathbf{y}$. You could probably just call this an approximation, but then we don't know how good the approximation is. Maybe I am missing something- please clarify.

7. Page 5, line 11: "two two".

8. Page 5, line 18: What do you mean by a "statistically consistent system"?

9. Page 5, line 21, Equation 8: What is $K$? I do not recall it having been defined previously.

---

## Author Response (AR2)

**Response to Referees' Comments**

**Peter Rayner**

**July 11, 2019**

I thank Amy Braverman and an anonymous referee for their further comments on the manuscript. the most important comment is that the paper should be more self-contained. Thus I have summarised some of the relevant sections from Rayner et al. (2018). Below I respond to reviewers' comments in detail. I have used a typewriter font for the reviewer's comment and Roman for my response

**Reviewer One**

**General Comments**

`. Page 2, lines 24 to 28: What does This required running optimized fluxes through the forward model used to generate the Jacobians have to do with challenging equal weighting? What is TM3?` I have expanded this explanation and explained the acronym.

`2. Page 3, Equation (1): Here is a case where you have referenced Rayner et. al. (2016). I find the material here impossible to understand without going back to that 2016 article, and even so, its not clear where this formula is coming from. In Rayner et. al. (2016), a similar version of current papers Equation (1) appears as Equation (2), but it doesnt look right to me: is it missing and integral? For the left-hand side to be p(x), you would have to integrate out all the other variables. You appear to be headed that way in Equation (1) of the current paper by integrating out yt, as if H() is a deterministic function of x. However, this is never stated, and is contrary to both the notation and treatment of H later.` This is a good point. I have expanded this section to summarise some of the relevant material from Rayner et al. (2018) and worked through the notation throughout.

`3. Its difficult to tell what the derivations in the first part of Section 2 are trying to show. It looks to me that you want to end up with Equation (3), which is an expression for 1`
`P(x, Hi|y), but this is only an intermediate step towards getting p(x|y): p(x|y) = P i p(x, Hi, y) p(y) = P i p(x|Hi,`

y) p(Hi, y) p(y) , = P i p(x|Hi, y) p(Hi|y) p(y) p(y) = X i
p(x|Hi, y) p(Hi|y). (1) Moreover, you want the expression
for p(x|y) to factor in such a way that it involves estimating
p(Hi|y) because those are the weights for the transport models,
and you are interested in those for their own sakes. I think
this argument could be made more clearly if you started Section
2 by stating that the ultimate goal is to obtain the moments
of p(x|y), which can be factored in different ways, and the
particular factorization above is the most informative because
it involves estimating the weights p(Hi|y). This is a good com-
ment and I have reordered the material to first explain the motivation. Here I had tried
to show that it arose from the hierarchical formalism but it can just as easily be stated
as a goal at the beginning.

4. In the footnote on page 3 you explain that Hi is intended
to be an indicator variable that really represents the index
into a set of transport models. You also say that H1, . .
. , HN are the Jacobians of those transport models (line 26).
Elsewhere, Hi is not bold (Equation (2)). These conventions
should all be described in the main text (no footnote) and
the meaning of bold versus non-bold should be clarified. I
suspect your use of non-bold Hi and non-bold x in Equation
(2) is because you are stating a generic result, and you are
not specifically referring to Hi and x used the rest of the
text in this section. Please explain that. I have now explicitly
described the move from potentially nonlinear $H$ to linear $\mathbf{H}$. The notation follows
from Rayner et al. (2018). I have also had someone else check the copy for font errors.

5. Line 27, page 3: Please define yO. I get that it is
the mean of the random vector that represents the observations,
but is it different that yt? It probably could be, but are
you making any assumptions about that? Also, here you treat
xb as the mean of the Gaussian distribution of the random variable
x, but Equation (1) treats it like a random variable (p(x|xb)).
Of course, it is possible that it could be both if the model
was hierarchical and specified a prior distribution on xb,
but if thats the case it should be stated. I suspect that
this is really just notation given that you write, G(x|, C)
on line 28 (if C is bold, then should also be bold). Finally,
the expression uncertainty covariance is somewhat confusing,
at least to me: should it just be covariance? Most of this I agree
and have implemented. I would rather, however, not drop the descriptor "uncertainty"
from covariance. There is an unfortunate habit of people confusing signal and uncer-
tainty covariances in this field and I would rather keep it clear in this paper, especially
as later I will move between the two.

6. Lines 2930 on page 3 and Equation (3): I dont understand
why this is here, but perhaps that is because my understanding
of what you are trying to do relies on expressions I wrote

above for item 3 (my Equation (1)).  The final expression for
p(x|y) there is already in terms of p(x|Hi, y).  You then write,
Thus our posterior for the ensemble is a mixture of Gaussians...,
which I agree with.  We both have p(Hi|y) (I note that you
have now switched to using capital P for probability instead
of p used earlier its a minor thing, but it would be better
to be consistent), and the remaining term I call p(x|Hi, y)
and you call G(x|xa i , Ai).  It might be helpful to clarify
this correspondence in the text since it ties back to the ultimate
objective of expressing the posterior p(x|y) in a special way
that admits the mixture of Gaussians representation. This section has been reordered in line with a previous comment which hopefully makes the development clearer. The Gaussian arises because that is the solution for the Bayesian problem for a single observation operator, I have now made this clearer. I have also proofread for things like capitalisation more carefully.

    7.  Lines 23, page 4:  When you say, As usual with a joint
PDF we obtain the marginal probability for a variable by integrating
over all others, to what are you referring?  Are you justifying
Equation (4)? This point has now been moved earlier but I am a little unsure what the reviewer refers to here, this seems a conventional statement about marginal probabilities.

    8.  Equation (4):  There are a few things about this that
need to be addressed or explained.  First, you stated earlier
in the footnote on page 3 that Hi is a stand-in for an index
random variable that distinguishes between transport models,
but you use Hi anyway to remind the reader to what this index
refers.  If that remains true, then Hi is a discrete variable
here, not a continuous one.  If thats the case, then P(Hi)
is not Gaussian, and I dont think the right-hand-side of the
equation makes sense.  In Michalak et. al. (2005), the target
of inference is  which is a vector of continuously- valued
variance parameters, so it makes sense there.  I think what
you are trying to do with this expression is to obtain the
set weights associated with models represented by Hi as in
Raftery et. al., (2005) which you cite.  Alternatively, maybe
you have changed the notation implicitly to treat Hi (or more
properly vec(Hi)) as a Gaussian random vector.  If so, please
explain. Indeed, I have fixed the inconsistency between using $\mathbf{H}_i$ or $i$. I hope the new, more careful development is clearer. But yes, the variable is discrete and I am following Raftery et al. (2005). I wonder if part of the confusion turns on what is variable and what fixed in this equation. I have now commented on this explicitly.

    I have divided the following into several subpoints. 9.  Lines 710, page
4:  Several issues here.  First, the words of the first sentence
in Section 2.1 provide an example of where xb is now discussed
as if it were a random variable rather a parameter (in contrast
to its use earlier in the paper).  Is xb a parameter of the

`prior distribution of x or is it a random draw from that distribution?`
`Only in the latter case does the notion of independence from`
`y make sense.` This is correct, I meant the prior distribution for $\mathbf{x}$ and have corrected.

Second, Equation (4) as stated is not the probability of simulating the observations (y); it is the probability of Hi given the observations. Should it be p(y—Hi, x)? My point here is that the two quantities are the same. Up to normalisation, $p(\mathbf{H}_i|\mathbf{y})$ turns out to be the PDF for the quantity $\mathbf{H}_i\mathbf{x} - \mathbf{y}$ evaluated at the point $\mathbf{H}_i\mathbf{x}^{\mathrm{b}} - \mathbf{y}^{\mathrm{o}}$. I have now developed this explicitly.

Third, I question assertion made in Michalak et. al. (2005), Section 6.4, Equation (4) that Equation (2) of that paper can be written, p(x) G(x xb , B) G(H(x) y, R). Equation (2) in Michalak et. al. (2005) is p(x) p(x—xb) p(yt—y) p(yt—H(x)). It appears to me that p(x—xb) (or p(x—xb) using the notation of the paper under review) is missing from the expression above. I don't agree, I think Michalak's $G(\mathbf{x} - \mathbf{x}^{\mathrm{b}}, \mathbf{B})$ (which I might write $G(\mathbf{x}, \mathbf{x}^{\mathrm{b}}, \mathbf{B})$ is your $p(\mathbf{x}|\mathbf{x}^{\mathrm{b}})$. My more explicit treatment no longer refers to the Michalak result at this point however, so this disagreement is no longer relevant to the current paper.

Finally, also G(x xb, B) is ambiguous at best and nonsense at worst: do they mean G(x xb—xb, B) and G(H(x) y—H(x), R)? As I said above, I would not write the Gaussian this way. I also note generally that I have now acceded to the reviewer's implicit suggestion and listed explicit dependence on observations (or whatever else) to distinguish prior and posterior probabilities. It makes the notation a little clumsier but seems necessary to avoid serious confusion.

    `10. Lines 1719, page 4: P(Hi) appears several times in`
`this passage. Do you mean P(Hi|y)?` Yes, see previous comment.

    `11. Lines 2228, page 4: Whats the point of this second-to-last`
`paragraph of Section 2.1? Is it simply to draw a line between`
`the more familiar concept of 2 in the literature and the work`
`here? You do use it in the next paragraph (and in Section`
`2.2), so perhaps these should all be combined into one paragraph?`
`That would make it clear why 2 is being defined. Also, I dont`
`understand the calculation given in lines 2527.` I have moved the $\chi^2$ paragraph into the next section and expanded the point on inconsistency.

    `12. Section 2.2: The statement that neither AIC nor BIC`
`take account of different prior uncertainties among parameters`
`or different sensitivities of the observations to these parameters`
`is mysterious to me. That is certainly true, but thats not`
`their purpose. Since I am confused about what Hi means here`
`notationally, and that makes it hard to understand what you`
`are driving at.` This comment is a little difficult to interpret. Perhaps the reviewer thinks I am criticising the other criteria? I am not but pointing out that there is a relationship between them and where the difference lies. The second point I can't yet respond to, here I do use $P(\mathbf{H}_i|\mathbf{y})$ as requested so I'm unsure where the confusion arises.

**Reviewer Two**

**Major Comments**

1. Some notations in this paper are not very consistent. For example, at the beginning of Section 2, the author used p() for probability density function (PDF), but later on P() was used for PDF. In addition, for function Hi() and matrix Hi, it is better to add some notes to make a clear distinction. Last, the criterion L in Equation (5) is not italic, but later on it appears in italic font and hence can be a bit confusing. This was also noted by reviewer one. I have enlisted help with proofreading.

   2. The conditional densities in Equations (1) and (4) are also conditional on xb , and hence the author should mention xb is omitted for notation simplicity. Besides, is the prior mean xb treated as a fixed or random quantity in this paper? see point nine from reviewer one. I have now tried to add explicit dependence throughout.

   3. Page 4, Line 7, the author mentioned that Provided xb and y are independent, R + HiBHT i is the variance of the prior mismatch y Hixb ..., which seems to be inappropriate. This is because the matrix B is the covariance matrix of x, not of the prior mean xb Indeed, this was poorly expressed, also responded to at point nine from reviewer one.

   4. Page 6, in Figure 1, why the weight of model 3 is so small for the tuned case, compared with other two cases? This took some digging. The tuning procedure returned 1 for the weights for this model. For most other models it substantially reduced prior uncertainty so increasing the unnormalised weight. When we applied the normalisation criterion (sum to 1) model 3 was severely punished. Model 1 suffered a less extreme version of the same thing, again its prior uncertainty was reduced less than most. This is a curious enough fact that I have added it to the discussion of the figure.

   5. The author claims that Equation (7) is the variance of the ensemble, which seems to be incorrect. From the formulation, it seems to be the mean squared (prediction) error for x. I don't think so although it looks like the prediction error. I've not found a derivation of this so I include it here so the reviewer can check my algebra. I present the univariate version, the multivariate will undoubtedly follow with much unpleasant matrix algebra.

Define a Gaussian mixture PDF

$$P(x) = \sum_{i=1}^{N} w_i G(x, \mu_i, \sigma_i^2) \tag{1}$$

with $G(x, \mu_i, \sigma_i^2)$ a Gaussian with mean $\mu_i$ and variance $\sigma_i^2$. the mean of $P$ is given by

$$\mu = \int x P(x) dx = \sum_{i=1}^{N} w_i \int x G(x, \mu_i, \sigma_i^2) dx = \sum_{i=1}^{N} w_i \mu_i \tag{2}$$

The variance is given by

$$\text{var} = \int (x - \mu)^2 P(x) dx \tag{3}$$

$$= \sum_{i=1}^{N} w_i \int (x - \mu)^2 G(x, \mu_i, \sigma_i^2) dx \tag{4}$$

$$\tag{5}$$

We add and subtract $\mu_i$ inside the bracketed term and expand to yield:

$$\text{var} = \sum_{i=1}^{N} w_i \Big[ \qquad\qquad\qquad\qquad = int(x - \mu_i)^2 G(x, \mu_i, \sigma_i^2) \tag{6}$$

$$+ 2(\mu - \mu_i) \int (x - \mu_i) G(x, \mu_i, \sigma_i^2) dx \tag{7}$$

$$+ \qquad\qquad\qquad\qquad \int (\mu - \mu_i)^2 G(x, \mu_i, \sigma_i^2) dx \Big] \tag{8}$$

The first integral in Eq. 6 is $\sigma_i^2$ by definition, the second integral is zero by antisymmetry of the integrand and the third integral is $(\mu_i - \mu)^2$, yielding the desired result.

    6.  Page 7, Figure 2:  The titles of boxplots are repeated
for each row but it is supposed that the results for all the
22 regions are reported.  The author should double check whether
this figure is correctly produced. Indeed it was not, corrected.
    7.  For Equations (8) and (9), it is better to give the
mathematical definition of the mean terms (e.g., the mean of
H(x)b i ); also the superscript a is missed in Equation (9).
Could the author provide more motivations for using Rprior
i,j and Rsample i,j ? I have added a separate equation for the mean before
Eq. 8. I don't think this is relevant for Eq. (9) since that uses the observations. Super-
script corrected. Most importantly I have added extra text on the motivation.
    8.  Page 9, Line 6:  The author pointed out that the residual
covariances have the largest values for a few terrestrially-influenced
stations such as Baltic Sea and so on.  A figure showing the
residual covariances can be added to support this claim. Done.
    9.  Page 10, for the section of computational aspects:  Provided
that R is a sparse matrix (e.g., diagonal), I think the computational
trick is to use a low-rank matrix to approximate HiBHT i ;
then we can resort to the Sherman-Woodbury-Morrison inversion
formula to compute the inverse of (HiBHT i + R) and the Sylvesters
theorem to compute its determinant (e.g, Cressie and Johannesson,
2008; Sang and Huang, 2012).  The author may add a bit more
details to make the computational strategy more clear. That
is a good strategy when one has a matrix representation for the model and when one
of the dimensions is reasonably small. Many problems do not meet these criteria so

we can only calculate matrix-vector products. The reviewer's case is common enough though so I have added it as an alternative.

10. Page 12, Figure 4: Similar to Figure 2, the results seem to be repeated and not all the regions statistics are reported. The author should double check whether the figure is correctly produced. fixed as above

**Minor Comments**

1. Page 1, Line 23, the right bracket should be removed. removed

2. Page 2, Line 12, discreet should be discrete. corrected

3. Page 3, Line 2: the in the standard data assimilation... should be capitalized. Similarly, Page 6, Line 11: the in the variance is calculated as should be capitalized. The author needs to double check whether there are similar typos in the paper. A hard one to pick up nonvisually that. Checked throughout

4. The author refers the Equation (1) but I do not see Equation (1) in the context.

5. Page 4, in the second and third paragraph, it seems that P(Hi) should be P(Hi|y). I have added these conditional expressions throughout.

6. Page 4, Line 23: ...2 is equal to the number of observations... should be ... the expected value of 2 is equal to the number of observations... corrected

7. Page 7, Line 1: The Superscripts * indicates we consider... should be The superscript * indicates we consider... corrected

8. Page 9, Line 6: Eq. 9 and Eq. 8 should be Eq. 8 and Eq. 9. corrected

9. Page 10: the math symbols, Xb and Xa should be xb and xa , respectively. More problems with capitalisation, corrected.

10. Page 11: in the caption of Figure 3, the author should give the full name of JIC. In fact I have stopped using the name so this caption has been rewritten.

**References**

[revised manuscript text omitted]

$$\underline{P}p(\underline{x}\mathbf{x}, H_i) = \underline{P}p(\underline{x}\mathbf{x}|H_i)\underline{P}p(H_i) \tag{3}$$
* * *
[1]The true target variable is $i$, the index variable on the set of observation operators but we will continue to use $H_i$ to make it clear to what this index refers.

 Combining Equation **??** and Equation 2 we obtain

$$p(\mathbf{x}, H_i|\mathbf{y}) = P(\mathbf{x}|\mathbf{y}, H_i)p(H_i|\mathbf{y}) \tag{4}$$

we see that the hierarchical and nonhierarchical PDFs differ only by the factor  $p(H_i|\mathbf{y})$ and we hence need to calculate this term.

5    We will develop the theory for the simplest linear Gaussian case. Here many of the resulting integrals have analytic solutions. The approach will hold for nonlinear observation operators provided they are approximately linear over enough of the support for the joint distribution of $\mathbf{x}$ and $\mathbf{y}$. The qualitative ranking of models is unlikely to be sensitive to weak nonlinearities since, as we shall see, the discrimination among models is strong.

We follow the notation of **?**. We switch from using a potentially nonlinear observation operator $H$ to a

10   linear one represented by the Jacobian $\mathbf{H}$. Take a collection of linear observation operators with Jacobians $\mathbf{H}_1 \ldots \mathbf{H}_N$, with prior probability for the  continuous target variables given by $G(\mathbf{x}|\mathbf{x}^{\mathrm{b}}, \mathbf{B})$ and  probability for the data given by $G(\mathbf{y}|\mathbf{y}^{\mathrm{o}}, \mathbf{R})$ where $G(\mathbf{x}|\mu, \mathbf{C})$ represents the Gaussian distribution of the variable $\mathbf{x}$  with mean $\mu$ and uncertainty covariance $\mathbf{C}$. $\mathbf{x}^{\mathrm{b}}$ is the prior or background with uncertainty covariance $\mathbf{B}$. $\mathbf{y}^{\mathrm{o}}$ is the observed value with uncertainty covariance $\mathbf{R}$ (**?**, Table 1).

[revised manuscript text omitted]

---

## Author Response (AR3)

**Response to Referees' Comments**

Peter Rayner

November 16, 2019

I again thank the reviewers for their comments. I address them pointwise below, using typewriter font for the reviewer comment and Roman for my response

**Reviewer One**

**Major Comments**

`1.  In Equation (1), the author introduced the notation of observed observations, yo.  I am wondering whether the conditional probability p(y|yo) in the rhs of Equation (1) should be p(yo|y), which is the data model when y denotes the underlying true data process.  In the meanwhile, to be clearer, p(x, y) may be written as p(x, y|xb, yo).` I was asked in an earlier version of the paper to include the dependencies of distributions explicitly. As a notation I did not like the change very much since, in my view, conditional probability should be reserved for the refinement of one PDF by conditioning with another. Here, parameters like $\mathbf{x}^b$ and $\mathbf{y}^o$ are not random variables but rather measured values which act as location parameters for a PDF. However I agree that the current state is an inconsistent halfway house so I agree with the reviewer's suggestion and have added $\mathbf{x}^b$ to the distribution.

`   2.  Page 3, Line 26:  the sentence Combing Equation 1 and Equation 3 is not necessary, since Equation (4) can be directly obtained based on the chain rule of probabilities.` Thank you, this is more direct and I have changed accordingly.

`   3.  Page 4, Line 16:  in Equation (6), the proportional symbol  should be =, since Equation (4) holds and p(x|y, Hi) = G(x, xa i , Ai) (Equation (5)).`  Agreed, this requires a few more changes too since the necessary rescaling of the directly calculated weights isn't automatic and must be accounted for somewhere. I have shifted the mention of this to the expression for $p(\mathbf{H}_i|\mathbf{y})$.

`   4.  Page 4, Line 20:  In Equation (7), p(Hi|y) should be p(Hi|yo ), since yo appears in the rhs of Equation (7),...` I disagree with this. I really mean the PDF for $\mathbf{H}_i$ conditioned on the distribution for $\mathbf{y}$ not just on the value of its location parameter $\mathbf{y}^o$.

`   5.  For Equation (7), I guess that the constant K should have a subscript i, since K depends on the prior distribution`

of different parameter models, p(Hi); also it is better to
mention that K is a constant in the context. No $K$ is a normal-
isation constant which arises from the fact that we must choose one of the models
so their probabilities must sum to 1. This normalisation constraint is now introduced
immediately after this equation.

6 In Figure 2 and Figure 5, I still noticed that the titles
of different boxes are repeated, and they are not the distinct
names for the 22 land-ocean regions of the TRANSCOM inter-comparison.
The author needs to double check whether these results are
correctly produced. Corrected. That is quite embarrassing. This time I have
checked the actual submitted version, not the almost-submitted version.

**Minor Comments**

1.  Page 1, Line 22:  The PDF needs to be explained. done.
   2.  Page 4, Line 9, the covariance B should have a bold
font. Done.
   3.  Page 4, Line 11, the brackets of the reference Rayner
et al., 2018, Section 6.4 should be removed. Done.
   4.  Page 5, Line 26:  The AIC should be 2M + 2 with the
penalty factor 2. Corrected.
   5.  Page 9, Line 29:  In Equation (14), the second (Hxb
j  b j) should have a transpose, (Hxb j  b j)T ; similarly,
in Equation (15), the second (Hxa j  yi) should be (Hxa j  yi)T
Corrected.

**Reviewer Two**

**Specific Comments**

1.  Page 3, lines 10 to 14 and Equation (1):  The notation
here is (still) ambiguous, and I do not follow how the left
and right sides of Equation (1) are equal.  You refer to xb
as the background, which I now understand to be the mean of
the prior distribution of x.  The prior variance is not specified,
so I am not sure how to interpret p(x|xb).  You state elsewhere
that Gaussian distributions are assumed so the variance would
need to be specified.  Same issue for p(y|yo) and p(y|H(x)).
You say, y represents the observations, but then o represents
the observed value.  So what is the difference between y and
yo?  My interpretation of Equation (1) as it is written is,
(equations deleted) This is as far as I can get without clarification
on y and yo.  In any case, it is not obvious to me why one
would take the product of these conditional distributions to
obtain p(x, y).  I dont think it would be obvious to a general

`reader either.` I think this points out nicely the problem i mentioned in response to the first point from Reviewer One. In general I do not think it wise to use the language of conditional probabilities to express the functional dependence of a PDF on its parameters. It leads to the kind of confusion we see here over what is and is not a random variable. Rather than restart that discussion I have added some explanatory text around Equation 1 to clarify.

`2.  Page 4, lines 7 to 9.  Here the prior covariance matrices of the distributions p(x|xb) and p(y|yo) are finally defined as B and R, respectively.  On line 9, it says yo is the observed value with uncertainty covariance R, but if yo is the mean of the prior distribution y, then R should be the covariance matrix of the prior distribution of y.  There is nothing in your model about yo being a random variable (this would imply a hierarchical model) up to this point.  Did you mean to say, y is the observed value with uncertainty covariance R?` This is right, I have clarified the language.

`3.  Page 4, line 16, Equation 6:  shouldnt this be an equality?` yes, corrected.

`4.  Page 4, line 17, This is nit-picking at some level, but you write Thus, p(x, Hi|y) is a sum of Gaussian Distributions... Technically, its not the distribution that is a sum of Gaussians, its the random variable that has a distribution that is a Gaussian mixture.` I do not quite understand this. Perhaps the reviewer is saying that since the target variable $\mathbf{H}_i$ is discrete then $p(\mathbf{x}, \mathbf{H}_i)$ cannot be though $p(\mathbf{x})$ will be once we integrate over $i$? I have assumed that is the problem and shifted the introduction of the term Gaussian mixture to Section 3.2.

`5.  Page 4, line 20, Equation 7:  Shouldnt the left-hand side of Equation 7 be p(Hi|yo )?  There is no term involving y on the right.  Also, I think you are missing a comma at the end of line 19, and a period at the end of Equation 7.` No, we are conditioning the PDF for $\mathbf{H}_i$ on the random variable for the observations $\mathbf{y}$. Once we have chosen a form for $P(\mathbf{y})$ (Gaussian in this case) we can write an explicit expression in terms of its parameters $\mathbf{y}^\circ$ and $\mathbf{R}$. I have added a colon on line 19 but will leave punctuation of equations to copy-editors.

`6.  Page 4, line 23:  You say Provided p(x) (the prior distribution for x) and y are independent....  Another nit-pick:  its not the distributions that are independent, its the random variables. More importantly, how can x and y possibly be independent? Isnt y = H(x) given the last term on the right-hand side of Equation 1?  I dont have the Tarantola book handy, so I am not sure what the Jacobian rule of probabilities is (will attempt to look this up).  I think what you are trying to do here is to justify adding the variances and ignoring covariance between x and y.  You could probably just call this an approximation, but then we dont know how good the approximation is.  Maybe I am missing something- please clarify.` Three points here. I agree

with the first and have modified the text. For the independence of $\mathbf{x}$ and $\mathbf{y}$, note that we were referring to the prior so the rhs of Equation 1 is not relevant. The third point is the most important since it suggests we haven't sufficiently motivated the development. We have added some earlier text to explain why we are doing this.

7. `Page 5, line 11: two two.` corrected.

8. `Page 5, line 18: What do you mean by a statistically consistent system?` I mean a system where assumed variances and calculated residuals are consistent, I have rewritten accordingly.

9. `Page 5, line 21, Equation 8: What is K? I do not recall it having been defined previously.` It is defined as a normalisation in Equation 7.

[revised manuscript text omitted]
}^{\mathrm{prior}}_{i,j} = \overline{(\mathbf{H}\mathbf{x}^{\mathrm{b}}_i - \mu^{\mathrm{b}}_i)(\mathbf{H}\mathbf{x}^{\mathrm{b}}_j - \mu^{\mathrm{b}}_j)} \tag{14}$$

 where once again the average is over the ensemble of models and the subscripts index the observations.

The second approach is analysis of the posterior residuals. Desroziers et al. (2005) noted that the residuals must be consistent with the PDF assumed for the model-data mismatch, here described by $\mathbf{R}$. If this is not the case we need to make a correction to $\mathbf{R}$. Here again we have a range of choices. If we have enough data we can fit covariance models as functions of space and
25  time. We do not have enough data so we calculate directly the ensemble covariance of the residuals as

$$\mathbf{R}^{\mathrm{sample}}_{i,j} = \overline{(\mathbf{H}\mathbf{x}^{\mathrm{a}}_i - \mathbf{y}_i)(\mathbf{H}\mathbf{x}_j - \mathbf{y}_j)} \tag{15}$$

where the overbar denotes an average over the ensemble of models and their respective analyses and the indices $i$ and $j$ refer to observations. Descriptively $\mathbf{R}^{\mathrm{sample}}$ will be positive if, on average, models make errors of the same sign for observations
* * *
[2]Strictly speaking it is the model PDF from **?**, but we have combined model and data uncertainties following their Section 6.4

[Figure]

**Figure 3.** Assumed standard deviations, standard deviations taken from the diagonal of $\mathbf{R}^{\text{prior}}$ and $\mathbf{R}^{\text{sample}}$ for a representative subset of stations.

$i$ and $j$. Note that if the ensemble of models is smaller than the number of observations (usually the case) then both $\mathbf{R}^{\text{sample}}$

30 and $\mathbf{R}^{\text{prior}}$ are singular.  Neither $\mathbf{R}^{\text{prior}}$ nor $\mathbf{R}^{\text{sample}}$ capture observational error however. (**?**, 
[revised manuscript text omitted]